# New strategies for vertical transport in chemistry-transport models: application to the case of the Mount Etna eruption on March 18, 2012 with CHIMERE v2017r4

Mathieu Lachatre[1], Sylvain Mailler[1,2], Laurent Menut[1], Solène Turquety[1], Pasquale Sellitto[3], Henda Guermazi[3], Giuseppe Salerno[4], Tommaso Caltabiano[4], and Elisa Carboni[5]

[1]LMD/IPSL, École Polytechnique, Institut Polytechnique de Paris, ENS, PSL Université, Sorbonne Université, CNRS, Palaiseau, France.
[2]École des Ponts, Université Paris-Est, 77455 Champs-sur-Marne, France.
[3]Laboratoire Inter-Universitaire des Systèmes Atmosphériques (LISA), UMR CNRS 7583, CNRS, Université Paris Est Créteil et Université de Paris, Institut Pierre Simon Laplace, Créteil, France.
[4]Istituto Nazionale di Geofisica e Vulcanologia, Osservatorio Etneo, Catania, Italy.
[5]Rutherford Appleton Laboratory, Chilton, Didcot, OX11 0QX, Oxfordshire, UK.

**Correspondence:** Mathieu Lachatre (Mathieu.lachatre@lmd.polytechnique.fr)

**Abstract.** Excessive numerical diffusion is one of the major limitations in the representation of long-range transport by chemistry-transport models. In the present study, we focus on excessive diffusion in the vertical direction, which has been shown to be a major issue, and we explore three possible ways to address this problem: increase vertical resolution, use an advection scheme with antidiffusive properties, and represent more accurately the vertical wind. This study is done with the CHIMERE chemistry-transport model, for the March 18, 2012 eruption of Mount Etna, which has released about 3 kt of sulphur dioxide in the atmosphere into a plume that has been observed by satellite instruments (IASI and OMI) for several days. The change from the classical Van Leer (1977) scheme to the Després and Lagoutière (1999) antidiffusive scheme in the vertical direction has been shown to bring the largest improvement to model outputs in terms of preserving the thin plume emitted by the volcano. To a lesser extent, improved representation of the vertical wind field has also been shown to reduce plume dispersion. Both these changes help reducing vertical diffusion in the model as much as a brute-force approach (increasing vertical resolution).

## 1 Introduction

Among many other uses such as operational forecast of air quality, chemistry-transport models (CTM) have been used successfully in the past to represent long-range transport of polluted plumes from different types of sources (mineral dust, volcanic eruptions, biomass burning etc.). The CHIMERE [1] CTM (Mailler et al., 2017) has previously been used to assess the possible impact of Eyjafjallajökull's 2010 eruption on air quality (Colette et al., 2011). Ash transport from this eruption has been modelled and compared to LiDAR vertical profiles, showing that the CHIMERE model represents correctly the advection of this

---

[1]www.lmd.polytechnique.fr/chimere/, last consulted 08/28/19

volcanic plume from its source in Iceland to the LiDAR (Light Detection And Ranging) facility located in Palaiseau (France), thousands of kilometers away. The altitude, location and timing of the plume was represented correctly, but the authors have shown that their simulation presented a strongly overestimated vertical spread of the plume. Similar studies focusing on volcanic plume dispersion from Boichu et al. (2013) have highlighted overestimations of plume diffusion on 2010 Eyjafjallajökull.

Several parameters can influence the evolution of the modelled plume: the emission fluxes and time profile; the injection height and vertical profile; chemical processes involving the considered species; wind field; numerical advection schemes and vertical resolution. Boichu et al. (2015) have focused on volcanic plume dispersion sensitivity to injection altitude, combining CHIMERE and Atmospheric Sounding Interferometer instrument (IASI) for a Mount Etna - study case of an eruption of moderate intensity in April 2011. The eruption presented an emission profile centered at 7 000 m.a.s.l., with weaker emissions at

4 000 m.a.s.l.. These authors found a strong sensitivity of model outputs to the altitude of injection. In Mailler et al. (2017), advection of the volcanic $SO_2$ plume emitted by a major eruption of the Puyehue Cordó Caulle volcano (Chile) is simulated with the CHIMERE model and compared to satellite measurements and to analyses provided by Klüser et al. (2013). This work shows that, after about one week of simulation, circumpolar transport of the plume has been represented correctly and the final position of the leading edge of the plume is simulated in a reasonable way, but plume dilution is excessive compared to the

observed shape and concentration of the plume.

Most CTMs have been built as offline models forced by meteorological fields, in particular wind and air density, taken from a forcing meteorological model, typically global forecast data such as outputs from the IFS (Integrated Forecasting System) or GFS (Global Forecast System), data from operational forecast centers or data generated by the modellers themselves from a locally run meteorological model. These meteorological data, after interpolation in time and space onto the CTM grid, are

used to drive advection within the CTM. However, grid type, grid structure, transport schemes and time discretization are generally different in the CTM from their formulations in the forcing models, deriving into mass-wind inconsistencies: once interpolated onto the CTM grid, the mass and wind fields do not obey the continuity equation anymore (Jöckel et al., 2001). While theoretical pathways to mitigation of this problem exist (Jöckel et al., 2001), this problem has historically been solved practically in regional CTMs in a straightforward way as described in Emery et al. (2011): reconstructing the vertical wind from

the density field and the horizontal mass flux divergence in order to artificially enforce verification of the continuity equation to the expense of the realism of the vertical mass fluxes, in particular in the free troposphere. This approach is justified by the fact that, in the lowest atmospheric layers, the reconstructed vertical mass flux is not very different from the real mass flux from the forcing model, and that this explicitly resolved vertical transport is usually dominated by mixing inside the Planetary Boundary Layer (PBL). Therefore, it has long been thought that this approach generates little if any problem since the main purpose of

regional CTMs is to provide a reliable forecast of the concentration of pollutants within the PBL. This historic focus on the PBL has also led to the habit of chemistry-transport modellers to use very loose vertical resolution in the free troposphere. Emery et al. (2011) describes the side effects on such an approach in two of the most used CTMs (CAMx and CMAQ). They show that oversimplification of vertical transport in the free troposphere can not only be detrimental to the representation of transport in the free troposphere itself but also defeats its own purpose: focus on obtaining a correct representation of pollutant

concentrations in the PBL. They have shown that oversimplified representation of vertical transport and vertical mass fluxes in

the free troposphere spuriously increases vertical transport of stratospheric ozone into the troposphere, resulting into degraded scores for forecast and analysis of ozone concentration in the PBL, particularly over complex and elevated terrain (springtime in the United States, in the case of Emery et al., 2011). To solve this problem, the authors tried different approaches. While trying to reduce spurious vertical velocities by applying mass filters, smoothers/desmoother filters or divergence minimizers

to the forcing velocity field either brings little improvement to the issue or introduces numerical artefacts, improvement of the vertical transport scheme and increase in the vertical number of layers in the free troposphere did bring substantial improvement to the issue of excessive transport of stratospheric ozone into the stratosphere.

Apart from this wind-mass inconsistency issue, and more specifically for the representation of polluted plumes that are transported over a long range, Zhuang et al. (2018) have shown that correct representation of long-range transport of polluted

plumes in the free troposphere is severely limited by the insufficient vertical resolution. They show, through dimensional and theoretical arguments, that if $\Delta x$ is not at least several hundred times $\Delta z$, representation of long-range transport of plumes in the free troposphere is hindered primarily by this coarse vertical resolution. Increasing horizontal resolution in these conditions does not bring substantial added value in terms of reducing numerical diffusion of the plume. Since the $\frac{\Delta x}{\Delta z}$ in typical chemistry-transport models is around or below 20 (with a horizontal resolution of, e.g., $20\,\mathrm{km}$ for continental scale studies and vertical

resolution of, e.g., $1\,\mathrm{km}$), these authors claim that no major improvement will be reached in the representation of long-range transport plumes unless vertical resolution is refined drastically compared to current typical configurations. However, they do not examine the use of anti-dissipative transport schemes, which can be a possibility to reduce vertical diffusion without dramatically increasing vertical resolution. For a more detailed discussion of the theoretical ground of this relationship between horizontal and vertical discussion, the reader is referred to Zhuang et al. (2018).

In the present study, three options have been tested in terms of the accuracy of representation of the long-range advection of thin layers. One option is to choose the Després and Lagoutière (1999) anti-diffusive advection scheme for vertical transport, the second option is to use realistic vertical mass fluxes instead of reconstructed vertical mass fluxes, the third option being refinement of vertical resolution. We have chosen the March 18, 2012 Etna volcano eruption to evaluate the impact of these new strategies for vertical transport. This volcano is well monitored, thus allowing to gather detailed model inputs and correlative

data. This eruption has been relatively strong, so that the resulting volcanic plume has been distinctly observed and followed by satellite instruments, permitting comparison of modelled to observed plume at different stages of plume evolution over more that two days. Etna volcanic activity is monitored continuously to estimate $SO_2$ fluxes and plume injection height (Salerno et al., 2009; Mastin et al., 2009; Sellitto et al., 2016; Salerno et al., 2018). Volcanic gases and aerosols are also subject to numerous physical and chemical evolution processes, such as sulphate production or CCN activation, which have been recently studied

(Sellitto et al., 2017; Guermazi et al., 2019; Pianezze et al., 2019) but are not accounted for in this study.

The manuscript is structured in the following way: Material and method (section 2) presents the CHIMERE model configuration for these simulations, including a detailed presentation of the transport formulation and its discretization in the CHIMERE model, adaptation of the Després and Lagoutière (1999) scheme to the CHIMERE framework and presentation of the method for compensation of mass-wind inconsistencies that permits us to use realistic vertical mass fluxes instead of

reconstructed mass fluxes. Also in this section, we discuss the satellite data that we used as a comparison point for our model

outputs, the different settings of the performed sensitivity tests and the $SO_2$ emission fluxes that we use. In the results and discussion (section 3), eruption injection altitude impact on plume transport is first investigated and compared to the plume transport constructed from satellite based instruments. In addition, sensitivity to vertical profile of injection has been evaluated. Then, the dispersion and trajectory of the simulated plumes is discussed and compared to the available data, with a focus on

the impact of the various tested parameters on plume dispersion (vertical wind representation, vertical advection scheme and number of vertical levels).

## 2    Material and methods

### 2.1    CHIMERE simulations

Simulations have been performed using a development version of the CHIMERE model (v2017r4; Menut et al., 2013; Mailler

et al., 2017) including the new developments presented in section 2.2. The simulations have been performed with no chemistry, and an inert gaseous tracer with the molar mass of $SO_2$ has been emitted at the location of the Etna volcano, with fluxes and injection heights that are presented below. No boundary conditions were used for $SO_2$ in our simulations. Oxidation of $SO_2$ and subsequent formation of sulphate or sulfuric acid have not been represented. Simulations last 120 hours starting on March 18, 00 UTC. The CHIMERE model has been forced using WRFv.3.7.1 (Weather Research and Forecasting Skamarock et al.,

2008), with an update of the forcing meteorological variables every 20 minutes using the WRF-CHIMERE online simulation framework (Briant et al., 2017). The WRF model has been run with 33 vertical levels from surface to 55 hPa (28 levels are into 1013-150 hPa range). The horizontal grid is the same as the CHIMERE grid, with a 5 km resolution. The meteorological boundary conditions have been taken from the NCEP GFS dataset at $0.25°$ resolution (NCEP, 2015), also used for the spectral nudging of the WRF simulation. The CHIMERE simulation domain is identical to the WRF simulation domain, with 799 ×

399 cells at 5 km resolution. The geometry of the domain, which has a Lambert-conformal projection, is shown in Figure 1. The top of model is placed at 150 hPa, with either 20, 50 or 99 vertical layers to evaluate the impact of vertical resolution on the volcanic plume. Even though a higher model top would have been useful for the study of this plume, 150 hPa is a typical value of top of model for CTMs that do not include stratospheric chemistry as it is the case of the CHIMERE model. Also, this relatively low value for top of model permits to examine the question of spurious mass fluxes through the top of model which,

as found by Emery et al. (2011) is of relevance not only for long-range transport but also for ozone forecast to ground level.

The discretisation of the vertical levels is as described in Mailler et al. (2017), with vertical levels of exponentially increasing thickness from surface to 850 hPa, and evenly spaced (in pressure coordinates) from 850 hPa. The vertical coordinate depends on the ground-level pressure, with finer vertical levels over elevated ground. The reader is referred to Mailler et al. (2017) (Section 3.1) for the detailed description of the vertical discretization of the CHIMERE model.

Horizontal advection in the CHIMERE model has been represented using the classical Van Leer (1977) second-order slope-limited transport scheme.

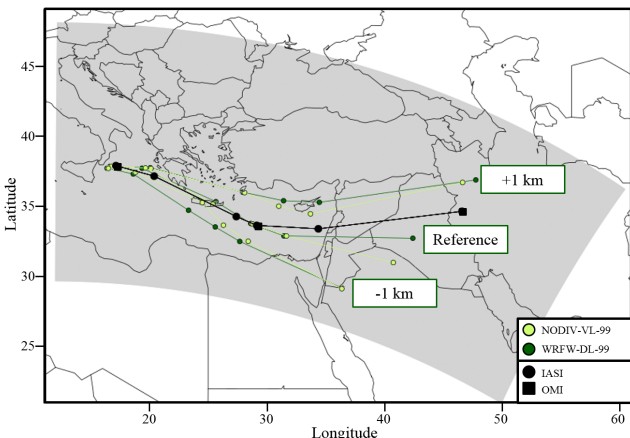

**Figure 1.** Satellite trajectory of the Etna volcanic plume (black line) built combining information from IASI and OMI instruments. CHIMERE simulated trajectory depending on $SO_2$ injection altitude of emissions (light and dark green lines - respectively NODIV-VL-99 and WRFW-DL-99). The grey area represents the CHIMERE simulation domain. White triangle indicates Mount Etna location.

## 2.2 Discretization of transport

Total concentration for all species together will be noted $C$ (number of gas particles per unit volume; molec.m$^{-3}$; corresponding to air density), concentration of a particular species $s$ will be noted $C_s$ (number of molecules of species $s$ per unit volume; molec..m$^{-3}$), mixing ratio for species $s$ will be noted $\alpha_s = \frac{C_s}{C}$. Continuity equation for the motion of air is as follows:

$$5 \quad \frac{\partial C}{\partial t} + \nabla(\mathbf{\Phi}) = 0, \tag{1}$$

where $\mathbf{\Phi} = C\mathbf{u}$ is the total flux of molecule number.

In the following equations, $i, j, k$ are the indices for the two horizontal directions and the vertical direction respectively, $F_{k+\frac{1}{2}}$ is the mass flux through the top of layer $k$, positively oriented for upward fluxes. Similarly the horizontal mass fluxes through lateral cell boundaries, positively oriented towards increasing $i$ and $j$ values respectively, are noted $F_{i+\frac{1}{2},j,k}$ and $F_{i,j,k}$.

10 The mass flux components need to verify the discretized form of Eq. 1:

$$\frac{\partial C_{i,j,k}}{\partial t} + \left(F_{i,j,k+\frac{1}{2}} - F_{i,j,k-\frac{1}{2}}\right) + \left(F_{i+\frac{1}{2},j,k} - F_{i-\frac{1}{2},j,k}\right) + \left(F_{i,j+\frac{1}{2},k} - F_{i,j-\frac{1}{2},k}\right) = 0 \tag{2}$$

The continuity equation for species $s$ is as follows:

$$\frac{\partial C_s}{\partial t} + \nabla(\mathbf{\Phi}_s) = 0 \tag{3}$$

or equivalently:

$$15 \quad \frac{\partial C_s}{\partial t} + \nabla(\alpha_s \mathbf{\Phi}) = 0 \tag{4}$$

In CHIMERE, Eq. 4 is discretized as:

$$\frac{\partial C_{s,i,j,k}}{\partial t} + \left( \bar{\alpha}_{i,j,k+\frac{1}{2}} F_{i,j,k+\frac{1}{2}} - \bar{\alpha}_{i,j,k-\frac{1}{2}} F_{i,j,k-\frac{1}{2}} \right) \tag{5}$$
$$+ \left( \bar{\alpha}_{i+\frac{1}{2},j,k} F_{i+\frac{1}{2},j,k} - \bar{\alpha}_{i-\frac{1}{2},j,k} F_{i-\frac{1}{2},j,k} \right)$$
$$+ \left( \bar{\alpha}_{i,j+\frac{1}{2},k} F_{i,j+\frac{1}{2},k} - \bar{\alpha}_{i,j-\frac{1}{2},k} F_{i,j-\frac{1}{2},k} \right) = 0$$

where $\bar{\alpha}$ are the reconstituted values of mixing ratio $\alpha$ on the facet indicated by the indices ($\bar{\alpha}_{i,j,k+\frac{1}{2}}$ for the top facet of cell $i,j,k$ etc.). If the mass flux values $F$ are such that Eq. 2 is exactly verified, then Eq. 5 ensures that the mixing ratios $\alpha_s$ are not affected by wind-mass discrepancies: in particular, a species with an initially uniform mixing ratio will maintain it after transport is applied. This is why it is so critical for chemistry-transport models to have Eq. 2 verified exactly.

Eq. 5 raise two important questions:

1. How to express the interpolated mixing ratios $\bar{\alpha}_s$, which is the task of the transport scheme ?

2. How to enforce exact verification of Eq. 2 to ensure the absence of mass-wind inconsistencies ?

### 2.2.1 Vertical wind strategy

In CHIMERE, as in most other chemistry-transport models (Emery et al., 2011), with the notable exception of chemistry-transport modules that are embedded within a meteorological model and use the same grid and time step as it is most notably the case of WRF-CHEM (Grell et al., 2005), the model does not have access to mass flux components $F_{i\pm\frac{1}{2},j\pm\frac{1}{2},k\pm\frac{1}{2}}$ and a density field $C$ that verify Eq. 2. This is a big problem when it comes to advecting species mixing ratios since as we have seen above, the fact that the CTM is able to transport species with Eq. 5 while maintaining uniformity of initially uniform mixing ratios critically depends on that property. Usually, chemistry-transport models rely on the typical outputs of meteorological models, namely the instantaneous values of winds at the meteorological model cell boundaries, and instantaneous values of density. From these variables, it is possible to evaluate the mass fluxes through the CTM cell boundaries, obtaining interpolated mass flux values that are close to the "real" mass flux values from the model. We will note these values $\widetilde{F}_{i\pm\frac{1}{2},i\pm\frac{1}{2},i\pm\frac{1}{2}}$, and $\widetilde{C}_{i,j,k}$.

With these interpolated values, and after discretization in time is also performed, Eq. 2 is not verified, and is turned into:

$$\frac{\partial \widetilde{C}_{i,j,k}}{\partial t} + \left( \widetilde{F}_{i,j,k+\frac{1}{2}} - \widetilde{F}_{i,j,k-\frac{1}{2}} \right) + \left( \widetilde{F}_{i+\frac{1}{2},j,k} - \widetilde{F}_{i-\frac{1}{2},j,k} \right) + \left( \widetilde{F}_{i,j+\frac{1}{2},k} - \widetilde{F}_{i,j-\frac{1}{2},k} \right) = -\varepsilon_{i,j,k}, \tag{6}$$

where $\varepsilon_{i,j,k}$ is a spurious matter creation term due to mass-wind inconsistencies in the interpolated density and mass-flux values from the meteorological model. $\varepsilon_{i,j,k}$ depends on the resolution of the meteorological model (which is identical for all our simulations), and on the resolution of the chemistry-transport model, so that this error term that essentially traduces divergence errors due to interpolation depends on the vertical resolution of the model. It is identical between simulations that

have the exact same number of domains. Choosing interpolation strategies that reduce this error term is a promising path to mitigating excessive vertical diffusion, as discussed in Emery et al. (2011), but is not investigated here.

As discussed above, the wind in CHIMERE is normally reconstructed from the bottom to the top of the model in order to prevent mass-wind inconsistency issues.

To enforce Eq. 2, reconstructed vertical fluxes $\overline{F}_{i,j,k+\frac{1}{2}}$ are produced from the following constraints:

$$\overline{F}_{i,j,\frac{1}{2}} = 0 \tag{7a}$$

$$\overline{F}_{i,j,k+\frac{1}{2}} = \overline{F}_{i,j,k-\frac{1}{2}} + \left( \widetilde{F}_{i-\frac{1}{2},j,k} - \widetilde{F}_{i+\frac{1}{2},j,k} \right) + \left( \widetilde{F}_{i,j-\frac{1}{2},k} - \widetilde{F}_{i,j+\frac{1}{2},k} \right) - \frac{\partial \widetilde{C}_{i,j,k}}{\partial t} \tag{7b}$$

Eq. 7a gives the boundary condition to vertical mass flux reconstruction (no incoming mass flux of air comes from the ground surface). Eq. 7b ensures that, in each CTM cell and over one CTM time step, Eq. 2 is strictly verified (in the form of 7b)

and Eq. 5 can be integrated using the interpolated horizontal fluxes $\widetilde{F}_{i\pm\frac{1}{2},j\pm\frac{1}{2},k}$ and the reconstructed vertical fluxes $\overline{F}_{i,j,k\pm\frac{1}{2}}$. This traditional approach will be labelled "NODIV" (for NO DIVergence) hereinafter.

In this study, we introduce and test a new approach that permits to bypass the need for a reconstructed vertical mass flux and work directly with the interpolated vertical mass fluxes $\widetilde{F}_{i,j,k+\frac{1}{2}}$ while still maintaining conservation of uniform mixing ratios. To explain this approach, we need to expand Eq. 5 as follows:

$$
\begin{aligned}
\quad & \frac{\partial C_{s,i,j,k}}{\partial t} + \left( \delta\bar{\alpha}_{i,j,k+\frac{1}{2}} \widetilde{F}_{i,j,k+\frac{1}{2}} - \delta\bar{\alpha}_{i,j,k-\frac{1}{2}} \widetilde{F}_{i,j,k-\frac{1}{2}} \right) \\
& + \left( \delta\bar{\alpha}_{i+\frac{1}{2},j,k} \widetilde{F}_{i+\frac{1}{2},j,k} - \delta\bar{\alpha}_{i-\frac{1}{2},j,k} \widetilde{F}_{i-\frac{1}{2},j,k} \right) \\
& + \left( \delta\bar{\alpha}_{i,j+\frac{1}{2},k} \widetilde{F}_{i,j+\frac{1}{2},k} - \delta\bar{\alpha}_{i,j-\frac{1}{2},k} \widetilde{F}_{i,j-\frac{1}{2},k} \right) \\
& + \alpha_{i,j,k} \left[ \left( \widetilde{F}_{i,j,k+\frac{1}{2}} - \widetilde{F}_{i,j,k-\frac{1}{2}} \right) + \left( \widetilde{F}_{i+\frac{1}{2},j,k} - \widetilde{F}_{i-\frac{1}{2},j,k} \right) + \left( \widetilde{F}_{i,j+\frac{1}{2},k} - \widetilde{F}_{i,j-\frac{1}{2},k} \right) \right] = 0,
\end{aligned} \tag{8}
$$

where $\delta\bar{\alpha}_{i,j,k+\frac{1}{2}} = \bar{\alpha}_{i,j,k+\frac{1}{2}} - \alpha_{i,j,k}$, and analogous definitions for the other $\delta\bar{\alpha}$ terms. Injecting Eq. 6 into Eq. 8 we obtain:

$$
\begin{aligned}
\quad & \frac{\partial \alpha_{s,i,j,k} \widetilde{C}_{i,j,k}}{\partial t} + \left( \delta\bar{\alpha}_{i,j,k+\frac{1}{2}} \widetilde{F}_{i,j,k+\frac{1}{2}} - \delta\bar{\alpha}_{i,j,k-\frac{1}{2}} \widetilde{F}_{i,j,k-\frac{1}{2}} \right) \\
& + \left( \delta\bar{\alpha}_{i+\frac{1}{2},j,k} \widetilde{F}_{i+\frac{1}{2},j,k} - \delta\bar{\alpha}_{i-\frac{1}{2},j,k} \widetilde{F}_{i-\frac{1}{2},j,k} \right) \\
& + \left( \delta\bar{\alpha}_{i,j+\frac{1}{2},k} \widetilde{F}_{i,j+\frac{1}{2},k} - \delta\bar{\alpha}_{i,j-\frac{1}{2},k} \widetilde{F}_{i,j-\frac{1}{2},k} \right) \\
& - \alpha_{s,i,j,k} \left[ \frac{\partial \widetilde{C}_{i,j,k}}{\partial t} + \varepsilon_{i,j,k} \right] = 0
\end{aligned} \tag{9}
$$

After simplification:

$$\widetilde{C}_{i,j,k}\frac{\partial \alpha_{s,i,j,k}}{\partial t}+\left(\delta\bar{\alpha}_{i,j,k+\frac{1}{2}}\widetilde{F}_{i,j,k+\frac{1}{2}}-\delta\bar{\alpha}_{i,j,k-\frac{1}{2}}\widetilde{F}_{i,j,k-\frac{1}{2}}\right) \tag{10}$$

$$+\left(\delta\bar{\alpha}_{i+\frac{1}{2},j,k}\widetilde{F}_{i+\frac{1}{2},j,k}-\delta\bar{\alpha}_{i-\frac{1}{2},j,k}\widetilde{F}_{i-\frac{1}{2},j,k}\right)$$

$$+\left(\delta\bar{\alpha}_{i,j+\frac{1}{2},k}\widetilde{F}_{i,j+\frac{1}{2},k}-\delta\bar{\alpha}_{i,j-\frac{1}{2},k}\widetilde{F}_{i,j-\frac{1}{2},k}\right)$$

$$-\alpha_{s,i,j,k}\varepsilon_{i,j,k}=0$$

From Eq. 10, it can be observed that if the mixing ratio $\alpha_{i,j,k}=\frac{C_{s,i,j,k}}{\widetilde{C}_{i,j,k}}$ is initially uniform, then all the $\delta\bar{\alpha}$ terms vanish, and mixing ratio uniformity will be maintained after integrating Eq. 8 if, and only if, the mass-wind inconsistency term $\varepsilon_{i,j,k}$ is zero. This is already well-known but with this formulation we can obtain a modified version of Eq. 5 that will enforce mixing ratio preservation even if the mass-wind inconsistency term $\varepsilon_{i,j,k}$ is not zero:

$$\frac{\partial C_{s,i,j,k}}{\partial t}+\left(\bar{\alpha}_{i,j,k+\frac{1}{2}}\widetilde{F}_{i,j,k+\frac{1}{2}}-\bar{\alpha}_{i,j,k-\frac{1}{2}}\widetilde{F}_{i,j,k-\frac{1}{2}}\right) \tag{11}$$

$$+\left(\bar{\alpha}_{i+\frac{1}{2},j,k}\widetilde{F}_{i+\frac{1}{2},j,k}-\bar{\alpha}_{i-\frac{1}{2},j,k}F_{i-\frac{1}{2},j,k}\right)$$

$$+\left(\bar{\alpha}_{i,j+\frac{1}{2},k}\widetilde{F}_{i,j+\frac{1}{2},k}-\bar{\alpha}_{i,j-\frac{1}{2},k}\widetilde{F}_{i,j-\frac{1}{2},k}\right)+\mathbf{C_{s,i,j,k}}\frac{\mathbf{\varepsilon_{i,j,k}}}{\widetilde{\mathbf{C}}_{\mathbf{i,j,k}}}=0$$

Eq. 11 will be solved in the simulation labeled WRFW, with mass fluxes directly interpolated from the meteorological model winds in the three directions. It must be noted that mass conservation is not enforced by this equation: the additional term $\mathbf{C_{s,i,j,k}}\frac{\mathbf{\varepsilon_{i,j,k}}}{\widetilde{\mathbf{C}}_{\mathbf{i,j,k}}}$ is an artificial mass production/loss term that breaks the conservation of total mass of species $s$ over the entire domain. If we summarize this part, the NODIV simulation (classical approach) enforces tracer mass conservation and tracer mixing ratio conservation. This is obtained to the expense of irrealistic vertical transport, since mass-wind consistency is, in this approach, enforced by artificially modifying the vertical mass fluxes. However, this reconstructed wind is significantly different from WRF input data while reaching the upper troposphere (Figure S1 in supplements), and this approach induces excessive transport across tropopause. Vertical wind distribution comparisons between WRFW and NODIV strategies (Figure S1) show that more vertical diffusion is expected in the NODIV strategy in the upper troposphere. The WRFW approach that we propose here, on the other hand, permits mixing ratio conservation and the use of realistic vertical mass fluxes, to the expense of mass conservation. While non-conservation of mass is obviously a significant drawback for a transport strategy, we will quantify this problem of non-conservation of mass, as well as the problems introduced by artificial reconstruction of vertical mass fluxes in the representation of vertical transport in the NODIV approach (Figure 3).

### 2.2.2 Vertical advection scheme

After discretizing the advection equation for species $s$ in the form of Eq. 5, the point of the vertical transport scheme is to estimate the reconstructed mixing ratios $\bar{\alpha}_{i,j,k+\frac{1}{2}}$, for k varying between 1 and the number of vertical levels $nz$. The most simple way of doing so is the Godunov donor-cell scheme, simply evaluating $\bar{\alpha}_{s,i,j,k+\frac{1}{2}}$ as:

$$\bar{\alpha}_{s,i,j,k+\frac{1}{2}} = \alpha_{s,i,j,k} \quad \text{if } F_{i,j,k+\frac{1}{2}} > 0 \tag{12}$$

$$\bar{\alpha}_{s,i,j,k+\frac{1}{2}} = \alpha_{s,i,j,k+1} \text{ if } F_{i,j,k+\frac{1}{2}} < 0 \tag{13}$$

This order-1 scheme is mass-conservative but extremely diffusive. It is therefore important to find more accurate ways to estimate $\bar{\alpha}_{s,i,j,k+\frac{1}{2}}$.

## 5 The Van Leer (1977) scheme

The second-order slope-limited scheme of Van Leer (1977) brought to our notations yields the following expression of $\bar{\alpha}_{s,k+\frac{1}{2}}$ (for $F_{i,j,k+\frac{1}{2}} > 0$).

$$\bar{\alpha}_{s,k+\frac{1}{2}} = \alpha_{s,k} + \frac{1-\nu}{2} \text{sign}\left(\alpha_{s,k+1} - \alpha_{s,k}\right) \text{Min}\left(\frac{1}{2}\left|\alpha_{s,k+1} - \alpha_{s,k-1}\right|, 2\left|\alpha_{s,k+1} - \alpha_{s,k}\right|, 2\left|\alpha_{s,k} - \alpha_{s,k-1}\right|\right), \tag{14}$$

where $\nu = \frac{F_{i,j,k+\frac{1}{2}}}{\rho_{i,j,k} V_{i,j,k}}$ is the Courant number for the donor cell $i, j, k$, $V_{i,j,k}$ being its volume: if $\nu > 1$, then more mass leaves
the cell than the mass that was initially present and the Courant-Friedrichs-Lewy condition is violated, yielding numerical instability. Eq. 15 is not applied in the case of a local extremum $((\alpha_{s,k} - \alpha_{s,k-1})(\alpha_{s,k+1} - \alpha_{s,k}) \leq 0)$. In this case, $\bar{\alpha}_{s,k+\frac{1}{2}} = \alpha_{s,k}$ is imposed and the scheme falls back to the simple Godunov donor-cell formulation (Eq. 12). This order-2 scheme has been used for decades in chemistry-transport modelling, being a good tradeoff between reasonably weak diffusion, at least compared to more simple schemes such as the Godunov donor-cell scheme, computationally cheaper than higher-order schemes such as
the Piecewise Parabolic Method (Colella and Woodward, 1984).

## The Després and Lagoutière (1999) scheme

The scheme of Després and Lagoutière (1999) is defined by their equations 2 to 4. If $F_{i,j,k+\frac{1}{2}} > 0$, these equations brought to our notations, adapted to the flux form of Eq.5 and ignoring the $i, j$ indices to shorten the expression, give:

$$\bar{\alpha}_{s,k+\frac{1}{2}} = \alpha_{s,k} + \frac{1-\nu}{2} \text{Max}\left[0, \text{Min}\left(\frac{2}{\nu}\frac{\alpha_{s,k} - \alpha_{s,k-1}}{\alpha_{s,k+1} - \alpha_{s,k}}, \frac{2}{1-\nu}\right)\right] \times \left(\alpha_{s,k+1} - \alpha_{s,k}\right), \tag{15}$$

with the same notations as for the Van Leer (1977) scheme (above). As above, Eq. 15 is not applied in the case of a local extremum $((\alpha_{s,k} - \alpha_{s,k-1})(\alpha_{s,k+1} - \alpha_{s,k}) \leq 0)$. In this case, $\bar{\alpha}_{s,k+\frac{1}{2}} = \alpha_{s,k}$ is imposed and the scheme falls back to the simple Godunov donor-cell formulation (Eq. 12). As stated by its authors, this scheme is antidiffusive. Unlike other schemes such as the Van Leer (1977) scheme described above, two unusual choices have been made by the authors in order to minimize diffusion by the advection scheme:

– Their scheme is accurate only to the first order

**Table 1.** List of the various model parameters tested, allowing to perform a total of 12 distinct simulations.

| Vertical levels | Vertical transport scheme | Vertical wind strategy |
|:---:|:---:|:---:|
| 20 | VL | NODIV |
| 50 | DL | WRFW |
| 99 | | |

    – The scheme is linearly unstable, but non-linearly stable (their Theorem 1)

The idea of the authors has been to make the interpolated value $\bar{\alpha}_{s,k+\frac{1}{2}}$ as close as possible to the downstream value ($\alpha_{s,k+1}$ if the flux is upward). This property is desirable because it is the key property in order to reduce numerical diffusion as much as mathematically possible while still maintaining the scheme stability. The authors present 1d case-studies with their scheme obtaining extremely interesting results: fields that are initially concentrated on one single cell do not occupy more than 3 cells even after a long advection time (their Figure 2), sharp gradients are very well preserved (their Figure 1), and, more unexpectedly due to its antidiffusive character, the scheme also performs well in maintaining the shape of concentration fields with an initially smooth concentration gradient. After extensive testing, these authors also suggest (their Conjecture 1) that convergence of the simulated values towards exact values occur even if the time step is reduced before the space step: in simpler terms, this means that the scheme performs very well even at small CFL values, a property that is not shared by most advection schemes.

### 2.2.3 Model parameters tested

The various possible parameter combinations between the vertical flux (NODIV for reconstructed vertical fluxes, WRFW for interpolated vertical mass flux), the vertical transport scheme (VL for Van Leer (1977), DL for Després and Lagoutière (1999)) and the number of vertical levels (20, 50, 99) are summarized in Table 1. Following all the possible combinations between these parameters, 12 simulations have been performed.

### 2.3 SO$_2$ emissions from the March 18 2012 eruption of Mount Etna

The time and altitude profiles for injection of SO$_2$ into the atmosphere (Table 2) have been obtained using SO$_2$ flux measurement data from the ground-based DOAS FLAME (Differential Optical Absorption Spectroscopy FLux Automatic MEasurements) scanning network (e.g. Salerno et al., 2018). This method accurately measures SO$_2$ fluxes during passive degassing and effusive and explosive eruptive activity using a plume height inverted by an empirical relationship between plume height and wind speed (Salerno et al., 2009). In explosive paroxysmal events, as in the case in this study, the plume is ejected to higher altitudes and this linear height-wind relationship can not be used. Mass flux is retrieved in post-processing using the plume height estimated by visual camera and/or satellite retrieval.

**Table 2.** SO$_2$ hourly flux (kg.s$^{-1}$) estimates used as input for the CHIMERE model.

| date, time | SO$_2$ flux (kg.s$^{-1}$) | injection height (m.a.s.l) |
|---|---|---|
| 18/03/2012, 06 UTC | 12.36 | 4500 |
| 18/03/2012, 07 UTC | 9.42 | 6500 |
| 18/03/2012, 08 UTC | 466.87 | 12000 |
| 18/03/2012, 09 UTC | 276.09 | 12000 |
| 18/03/2012, 10 UTC | 31.95 | 6000 |
| 18/03/2012, 11 UTC | 3.73 | 4500 |
| 18/03/2012, 12 UTC | 4.24 | 4500 |
| 18/03/2012, 13 UTC | 5.05 | 4500 |
| 18/03/2012, 14 UTC | 4.12 | 4500 |
| 18/03/2012, 15 UTC | 4.38 | 4500 |

**Table 3.** IASI and OMI soundings list. $\lambda_{OBS,i}$ (longitude) and $\Phi_{OBS,i}$ (latitude) represent plume's column with the highest SO$_2$ content coordinates. $\lambda_{thr,i}$ is the limit longitude that has been set as limit between the Eastern and Western plumes in section 3.4.

| date, time | Instrument | sounding number, $i$ | $\lambda_{OBS,i}$ | $\Phi_{OBS,i}$ | $\lambda_{thr,i}$ | Figure |
|---|---|---|---|---|---|---|
| 18/03/2012, 12 UTC | OMI | 1 | 17°07′ | 37°52′ | | |
| 18/03/2012, 17 UTC | IASI | 2 | 20°23′ | 37°08′ | 17°45′ | Figure S2 |
| 19/03/2012, 06 UTC | IASI | 3 | 27°23′ | 34°15′ | 19°30′ | Figure 2 |
| 19/03/2012, 12 UTC | OMI | 4 | 29°07′ | 33°37′ | | Figure 2 |
| 19/03/2012, 17 UTC | IASI | 5 | 34°23′ | 33°23′ | 21°30′ | Figure S2 |
| 20/03/2012, 12 UTC | OMI | 6 | 46°37′ | 34°37′ | | |

On March 18 2012, between 06 UTC and 15 UTC, a total SO$_2$ emission of 2.94 kt has been reported by this method. 91 % of this mass has been released within 2 hours around 12 km of altitude. In CHIMERE, emissions have been distributed into a single model cell based on altitude injection. Two alternates cases have been tested, considering -1 km and +1 km for all of the injection heights defined in Table 2.

## 2.4 IASI and OMI instruments

To evaluate the numerical parameters tested in our simulations, satellite based information were used to evaluate the SO$_2$ plume transport and vertical distribution. SO$_2$ column observations are provided by the Infrared Atmospheric Sounding Interferometer instrument (IASI) on board the Metop-A European satellite and the Ozone Monitoring Instrument (OMI) on board

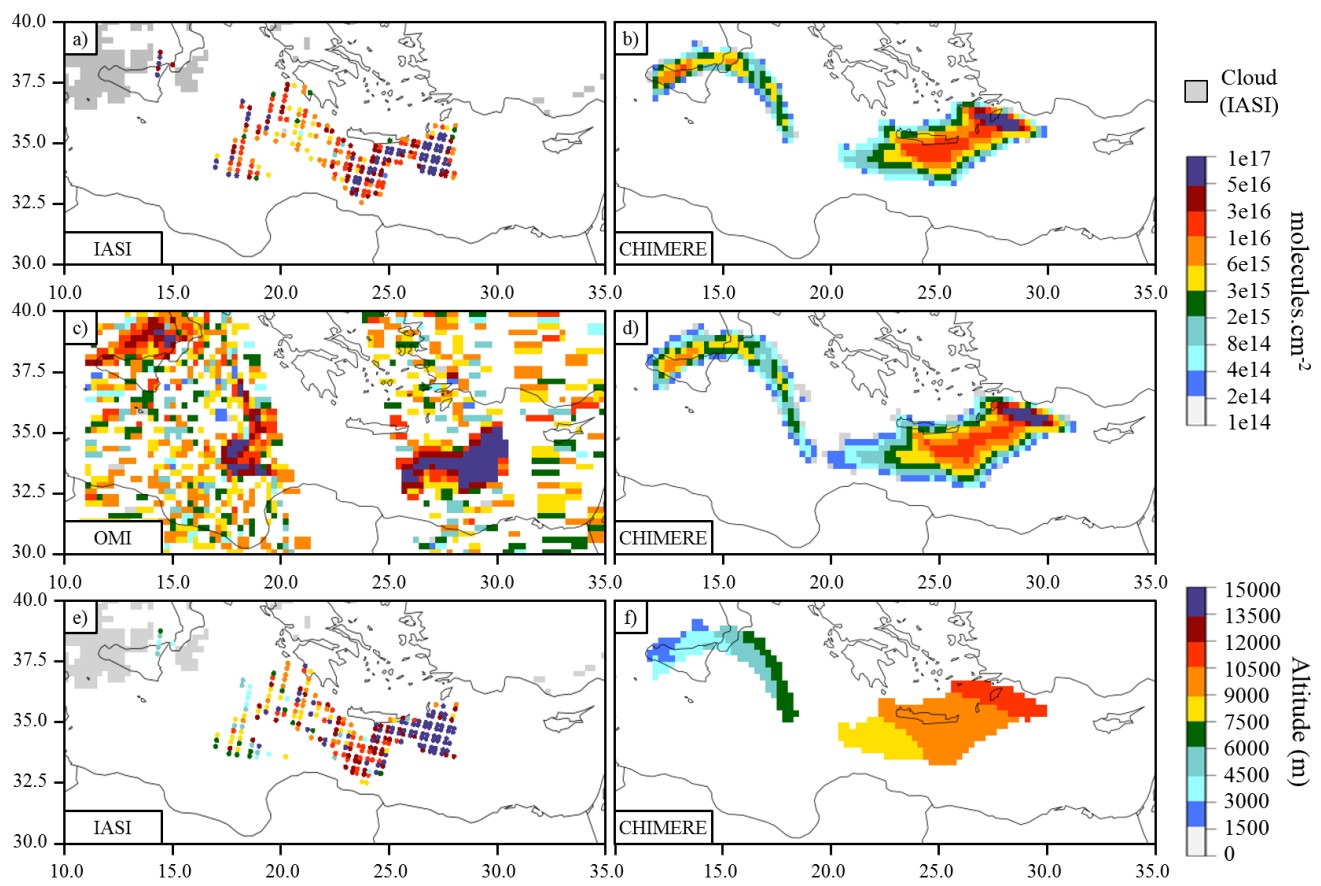

**Figure 2.** IASI SO$_2$ plume (a), OMI SO$_2$ plume (c), IASI and OMI soundings respectively for March 19 2012 06AM and 12AM UTC (3 and 4 in Table 3). CHIMERE SO$_2$ plume (b and d) in molecules.cm$^{-2}$ corresponding to IASI and OMI soundings. IASI SO$_2$ altitude (e) and CHIMERE SO$_2$ maximum concentrations' altitude (f) in meters March 19 2012 06AM UTC. In this example, CHIMERE simulation WRFW-DL-20 is displayed. CHIMERE and OMI data are represented with OMI's $0.25° \times 0.25°$ resolution grid. Clouds are based on Advanced Very High Resolution Radiometer (AVHRR) data for IASI. Logscale is used to better visualized CHIMERE simulations, but values under 3e15 ($\sim$0.1DU) are below satellite detection limits.

the Aura NASA satellite (Levelt et al., 2006; McCormick et al., 2013). IASI instrument is operating between 3.7 and 15.5 μm, including SO$_2$ $\nu_1$ (around 8.7 μm) and $\nu_3$ (7.3 μm) bands (Carboni et al., 2012). IASI scheme (Carboni et al., 2012, 2016) provides SO$_2$ total column content and plume altitude. This product is shown in Figure 2 together with OMI SO$_2$ columns and CHIMERE outputs. In our analysis we used IASI retrievals from March 18, 17UTC (Figure S2), March 19, 06UTC (Figure 2)

5 and March 19, 17UTC (Figure S3). OMI data are obtained from the NASA GIOVANNI platform[2]. OMI Data are provided

---

[2]https://giovanni.gsfc.nasa.gov/giovanni/, last consulted 08/29/19. Krotkov et al.

with a $0.25° \times 0.25°$ resolution and a daily coverage, at 12AM UTC for $18^{th}$, $19^{th}$ and $20^{th}$ of March over the studied area. All instruments soundings are resumed in Table 3.

## 3 Results and Discussion

### 3.1 Impact of injection altitude on plume transport

Alternative injection height scenarios have been tested, either lifting or lowering the injection height at all times by $1\,km$, thereby lifting maximum injection heights up to $13\,km$ (res. lowering it down to $11\,km$) instead of $12\,km$ in the reference simulation. These tests have shown that plume trajectories are strongly sensitive to this parameter (Figure 1), which is an effect of wind shear. For each satellite sounding, the coordinates of the model column with the strongest vertically integrated $SO_2$ content ($molecules_{SO_2}.cm^{-2}$) have been selected and considered as $SO_2$ plume centroids. Doing so with the available 3 IASI

soundings and 3 OMI soundings gives 6 points on the satellite-retrieved Etna plume trajectory, ranging from Sicily to Western Iran. The IASI/OMI centroids and constructed plume trajectory are displayed in black on Figure 4.

 Simulations and satellite plumes initial position do not correspond to Etna location because OMI first sounding is at $12\,UTC$ on March 18, 6 hours after the beginning of the eruption. Compared to the trajectory reconstituted from OMI and IASI observations, the plume injected at $11\,km$ seems to be transported too far towards the South, while the plume injected at $13\,km$

appears to be shifted to the North compared to observations. This observation is largely independent of the model parameters (NODIV-VL-99 and WRFW-DL-99 are shown on Figure 1), suggesting that the configuration with a maximum injection height at $12\,km$ is the best configuration. Therefore, only this choice for injection height will be retained for the rest of the study.

 In addition, sensitivity to injection vertical profile has been investigated with 3 options:

– Injection to a unique altitude

– Injection with a full width at half maximum of $100\,m$ (Boichu et al., 2015)

– Injection with a full width at half maximum of $300\,m$

The tests have been conducted on 20, 50 and 99 vertical levels resolution. These sensitivity tests have shown little differences between the various cases, even in 99 vertical levels resolution, with plumes close trajectories and vertical diffusion. Injection to a unique altitude - consequently in a unique cell - has been conserved and used to perform and evaluate the vertical diffusion

strategies.

### 3.2 Mass conservation

Figure 3 shows the evolution of the total mass of tracer inside the simulation domain as a function of time. Several features from this figures need to be commented.

 In the simulations with the reconstituted non-divergent wind field, substantial mass leak through the top of model can be

observed as soon as the injection starts in the 20-level simulation (in which injection is done in the highest model level): the

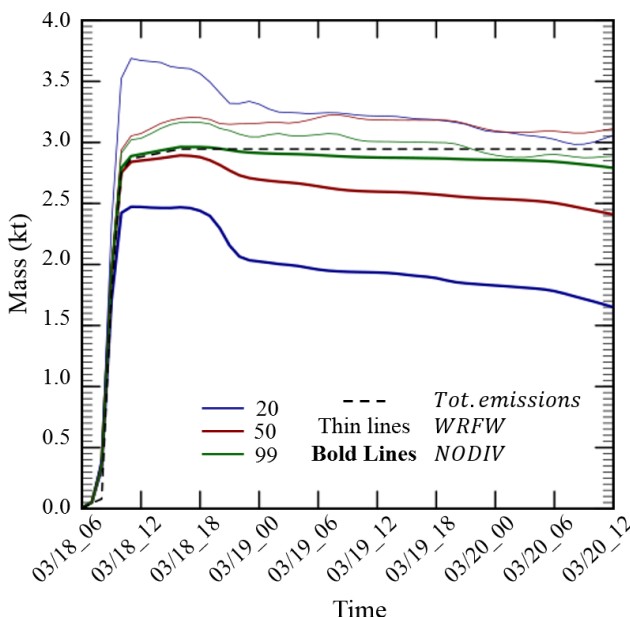

**Figure 3.** SO$_2$ mass evolution in model domain (kilotons). Line color indicates the vertical levels configuration, thickness indicates the vertical wind strategy considered. Dotted line represents the cumulated SO$_2$ mass emitted during Etna volcanic Eruption.

mass of tracer present in the domain never exceeds 85% of the emitted mass. For the simulation with 50 vertical levels, this phenomenon is also visible. Another strong episode of mass leak through model top occurs in the simulations with 20 and 50 vertical levels and with reconstructed wind fields from March 18, 18UTC to March 19, 00UTC. This episodes causes an additional drop in tracer mass of 20% in the simulation with 20 levels, 5% in the simulation with 50 vertical levels. This episode

of leak also affects the simulation with 20 vertical levels and with interpolated wind fields, reducing tracer mass concentration by about 10% from March 18, 18UTC to March 19, 00UTC. In these three simulations (20 and 50 levels with non-divergent winds, 20 levels with interpolated winds), a continuous decreasing trend in tracer mass is observed throughout the simulation. This drop is directly attributable to leak through model top since the tracer plume is far away from the horizontal boundaries of the domain.

As it could be expected, the simulations with the WRFW wind strategy, due to the additional term in Eq. 11, do not enforce mass conservation. In theory, the additional term, designed to ensure mixing ratio conservation in spite of mass-wind inconsistencies, can result into either an artificial increase or an artificial decrease in simulated tracer mass. In the three simulations that are shown on Figure 3, the amount of tracer present in the domain just after the end of the eruption overshoots the expected mass, by 20% in the simulation with 20 vertical levels, 10% in the two simulations with a larger number of levels. No physical

process can explain this overshoot, and it is directly attributable to the choice of lifting the mass conservation constraint in the formulation of transport in order to permit the use of a realistic wind field. If we take March 19, 00UTC as a reference time at which the eruption is terminated, the first strong event of leak through model top is terminated as well, we can observe that the

mass evolution in all three WRFW simulations undergoes small variations from one hour to the next but stay confined in very narrow ranges : 3.3 to 3 kt for the simulation with 20 vertical levels, with a decreasing trend attributable to leakage through model top, 3.1 to 3.25 for the simulation with 50 levels and 2.9 to 3.1 kt for the simulation with 99 vertical levels. The fact that these variations in total mass become marginal in this latter part of plume advection, when the plume is spread over a large geographic areas reflect the fact that numerical errors in the evaluation of divergence mechanically tend to compensate each other between neighbouring cells so that their global impact on a plume that is dispersed over many cells is small.

### 3.3 Horizontal transport evaluation with OMI and IASI instruments

In this section we aim to determine how the various parameters tested (Table 1) have influenced plume trajectory. Plume trajectory from simulations have been constructed following the same methodology as described above for satellite data, using the corresponding satellite sounding time step and retaining the coordinates of the model column with the strongest vertically integrated $SO_2$ content. The trajectories of all 12 simulations are shown in Figure 4a, with a color-code aimed at highilghting the impact of vertical levels number on the diffusion. Figure 4b allows to compare the influence of the vertical transport schemes (VL or DL), and finally Figure 4c allows to compare the various vertical wind strategies (NODIV or WRFW) influence.

It can be observed that for the various 20 vertical levels simulations, no significant differences between the vertical transport schemes nor vertical wind strategy are found, except for NODIV-VL-20 simulation which strongly diverges and split into 2 different plumes at OMI's last sounding. For 50 and 99 vertical levels simulations, more differences are found, mainly controlled by the choice of vertical transport schemes (VL or DL). As for NODIV-VL-20, NODIV-VL-50 simulation presents for OMI's last sounding a split in two distinct $SO_2$ plumes.

To conduct a more quantitative and synthetic analysis of the deviation between observations and model outputs, the geographic distance between satellite observation centroids and simulations centroids has been calculated for every sounding. This calculation provides for each simulation a satellite - model differences time series. Then, to better estimate the impact of the tested parameters, gap means have been calculated according to simulations parameters to evaluate separately the impact of each parameter choice on the accuracy of plume simulation. Results are displayed in Figure 5a. A general mean value for each time series is calculated and added on Figure 5a, last boxes. As expected, gaps between satellite and model centroids generally increase with time.

It can be seen that the DL vertical scheme has better agreement with the observations than the VL vertical scheme, with respectively a mean gap of $189\,km$ and $316\,km$ - with NODIV wind strategy option fixed. The WRFW wind strategy also shows better agreement with soundings than the NODIV strategy, with a mean gap of respectively $230\,km$ and $316\,km$ - with VL vertical scheme fixed.

To complete the centroids-gap analysis, agreement between satellite measurements and model simulations in the zonal and meridional displacements of the centroids has been calculated, as expressed in Equation 16 for a given simulation (SIM):

$$\Delta_{(i,i-1),\text{SIM}}(km) = R\sqrt{\cos^2\Phi\left(\Delta\lambda_{\text{SIM},(i,i-1)} - \Delta\lambda_{\text{OBS},(i,i-1)}\right)^2 + \left(\Delta\Phi_{\text{OBS},(i,i-1)} - \Delta\Phi_{\text{SIM},(i,i-1)}\right)^2}, \qquad (16)$$

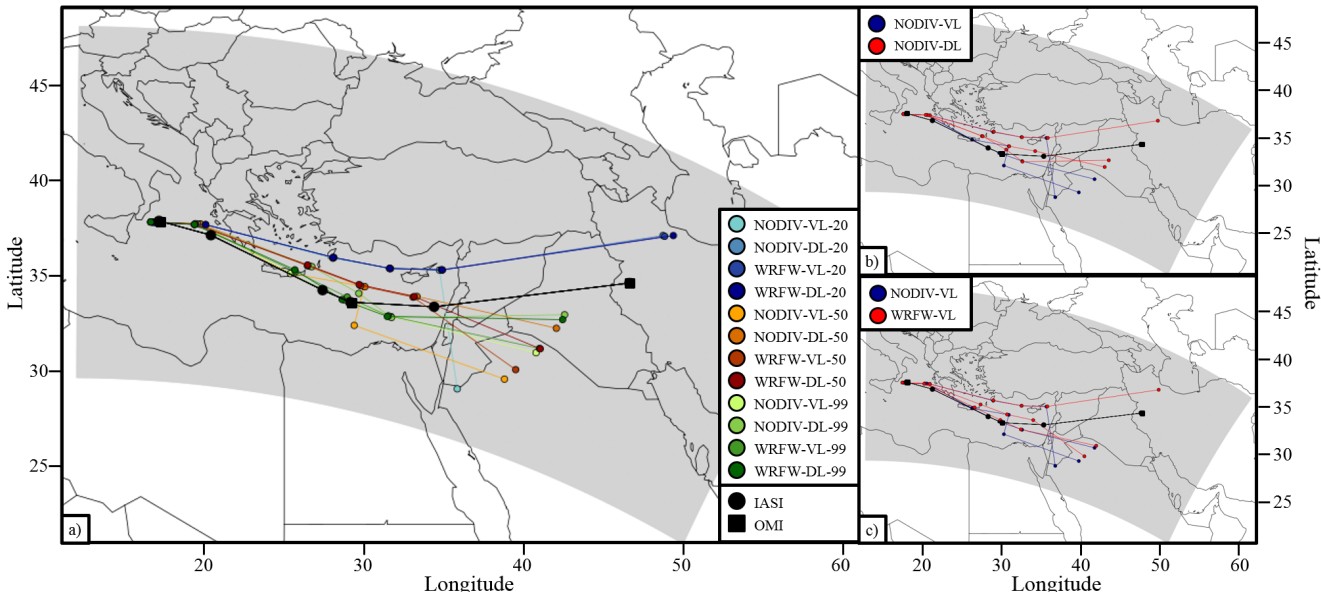

**Figure 4.** Etna volcanic plume transport over the Mediterranean sea after the March 18 2012 eruption. Satellite trajectory (black line) built combining IASI and OMI instruments information. a) Transport for all simulations, b) plume transport highlighted according to vertical scheme, c) plume transport highlighted according to vertical wind strategy. The points along the trajectories correspond to those listed in Table 3.

where

$$\Delta\lambda_{\mathrm{SIM},(i,i-1)} = \lambda_{\mathrm{SIM},i} - \lambda_{\mathrm{SIM},i-1}, \text{ and}$$

$$\Delta\lambda_{\mathrm{OBS},(i,i-1)} = \lambda_{\mathrm{OBS},i} - \lambda_{\mathrm{OBS},i-1}.$$

$i$ refers to sounding numbers (Table 3), $\lambda_{\mathrm{OBS},i}$ and $\Phi_{\mathrm{OBS},i}$ refer to the geographic coordinates of the observed centroid for

5 sounding $i$ at time $t_i$ (Table 3), $\lambda_{\mathrm{SIM},i}$ and $\Phi_{\mathrm{SIM},i}$ to the coordinates of the simulated centroid, and $\mathrm{R}$ is the Earth radius. In this case we focus on the displacement of the plume between two successive observation points. The intention of Eq. 16 is to build an index that not only qualifies the distance between the observed and modelled trajectories, but also the realism of the displacements followed between two successive satellite snapshots, giving penalties to simulations which would oscillate erratically around the observed trajectory and bonuses to simulations that would follow a realistic trajectory, but slightly

10 shifted towards either side. Results are displayed on Figure 5b, and again mean value for each time series has been calculated and displayed on Figure 5b last boxes. It can be observed, as in the previous indicator case, the DL vertical scheme and WRFW vertical wind strategy have brought better results than the respectively opposed VL and NODIV parameters.

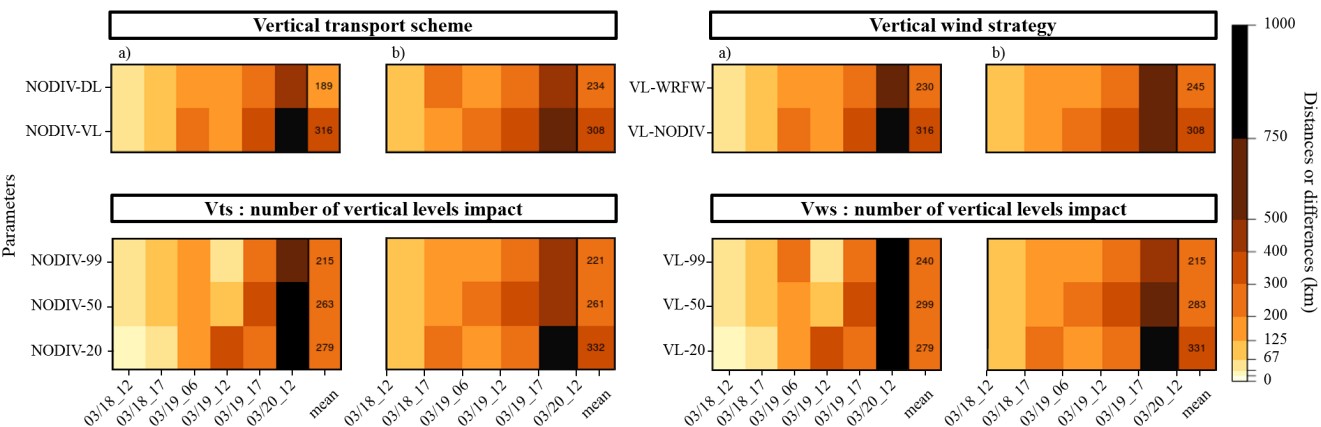

**Figure 5.** Left) Vertical transport scheme impact, Right) Vertical wind strategy impact. a) Gap between satellites and CHIMERE $SO_2$ plume centroids. b) Differences between $SO_2$ plume centroids trajectories. To produce this figure, differences (in km) between model and satellite plumes centroids are calculated for each simulation, then the impact of each parameter is evaluated by calculating the mean between simulation-satellite differences. For instance, "NODIV-DL" ($1^{st}$ line, left column) is the mean between "NODIV-DL-20", "NODIV-DL-50", "NODIV-DL-99". "NODIV-99" ($3^{rd}$ line, left column) is the mean between "NODIV-DL-99" and "NODIV-VL-99".

It also appears clearly for both criteria that the 99 vertical levels option shows the best results to both, centroids position and trajectory comparisons to the satellite than the 50 or 20 vertical levels options.

### 3.4 Comparison of the simulated plume vertical structure with IASI-retrieved structure

IASI observations also provide the estimated altitude of the $SO_2$ plume for each pixel with a valid $SO_2$ retrieval (Figure 2e),
along with the corresponding uncertainty. For each of the three available IASI soundings (Numbers 2, 3 and 5 in Table 3), we have extracted the plume altitude and its associated uncertainty for the pixel with the highest content of $SO_2$. Comparison of this plume altitude with the same calculation made on the plume simulated by CHIMERE is shown on Figure S4 (in supplements). It can be observed that for the IASI first available sounding, soon after the Etna eruption, concentration maximum altitude for CHIMERE is consistent with IASI altitude and is found within IASI uncertainties ($12\,100\,\text{m} \pm 900\,\text{m}$): Along with the
trajectories shown on Figure 1, this is an indication that the highest emission altitude at $12\,\text{km}$ in Tab. 2 is realistic.

In the IASI dataset, a bimodal altitude distribution is observed, indicating coexistence of two separated sub-plumes during this eruption: one located to the East at higher altitude, another one located to the West at lower altitude (Figure 2), which has also been observed in AEROIASI-sulphates soundings in (Sellitto et al., 2017; Guermazi et al., 2019). This separation is due to the sharp separation between emissions at very high altitudes ($\simeq 12\,\text{km}$) and emissions below $\simeq 7,5\,km$ (Tab. 2) and
to the fact that at the Etna latitude wind shear is generally strong with steady westerly winds in the higher troposphere and more variable winds in the lower troposphere. Since most of $SO_2$ is emitted around $\simeq 12\,\text{km}$ (Table 2), most of $SO_2$ mass is found in the Eastern plume. From each available IASI observation of the plume (soundings 2, 3 and 5 in Tab. 2), a transition

longitude that separates the western plume from the eastern plume has been identified. These longitudes are given in Tab. 2 and can be compared to Figures S2 and S3. The same longitudes have been used to separate the Eastern and Western plume in the CHIMERE simulations. Figure 6 compares the altitude distribution between the CHIMERE simulations and IASI retrievals - 20 vertical levels simulations have been removed, because the coarse altitude resolution does not permit a useful representation of the maximum concentration's altitude distribution (see Figure S4, in supplements). Altitude distribution median values are extremely close for both plumes between CHIMERE simulations and IASI soundings. The various parameters tested have not significantly changed the altitude distribution median but have impacted altitude distributions' widths, which have been slightly tightened for 99 vertical levels, DL vertical transport scheme and WRFW vertical wind strategy.

The IASI dataset also provides error-range estimates along with the retrieved plume altitude. These error-range estimates have a median of around $1000\,\mathrm{m}$ in the western plume and $5000\,\mathrm{m}$ in the eastern plume, which is much higher aloft. These uncertainties help to understand the wide distribution obtained from satellite. It is also worth noting that this dataset provides plume altitude but does not provide an information on plume thickness. Therefore, comparison between the left and right panels in Figure 2 does not represent the compared plume thickness between model and observation, but the compared variability of plume height. Unfortunately, due to the relatively large ucertainties affecting the retrieved altitudes, no conclusion can be made on this point either. With all these limitations, Figure 2 prove that model simulations represent the general structure of the plume, with an elevated eastern plume and a low western plume, and that the median altitudes of both these plumes are very comparable to the median of the satellite-provided altitudes.

## 3.5   Impact of model configuration on $SO_2$ vertical diffusion

To evaluate directly the impact of the various model configurations on $SO_2$ vertical diffusion, time evolution of the $SO_2$ vertical profile for the model column with the strongest $SO_2$ content at each hourly model output steps are shown on Figure 7, showing that:

- vertical levels are clearly insufficient to reproduce correctly even the main features of the plume advection in this case: no evolution of the plume altitude can be seen at such a coarse vertical resolution, and the plume seems to be strongly leaking through the top of model, which is not the case with 50 or 99 vertical levels

- The simulation that diffuses less the plume is the WRFW-DL-99

- Using an antidiffusive advection scheme permits to reduce diffusion almost as strongly as increasing the number of levels: for example, simulation NODIV-DL-50 preserves the maximum concentration of $SO_2$ in the plume and the thin plume structure as well as simulation NODIV-VL-99, with a calculation cost divided by two due to the reduced number of vertical levels. Therefore, the use of an antidiffusive advection scheme is a very attractive means of reducing numerical diffusion on in the vertical direction without increasing the computational cost of the simulation.

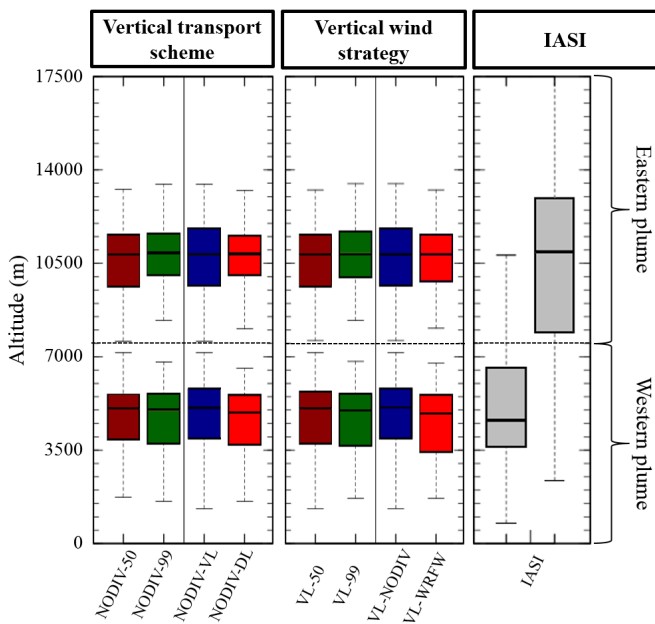

**Figure 6.** SO$_2$ plume maximum concentration's altitude distributions for IASI measurements and CHIMERE model simulations for the different configurations tested. Brackets correspond to distribution's $10^{th}$ and $90^{th}$ percentiles

- – Using the realistic vertical wind WRFW instead of the vertically reconstructed NODIV wind also reduces numerical diffusion and avoids intermittent leaks of the tracer through the upper model boundary, as can be seen by comparison between the NODIV-VL-20 and WRFW-VL-20 simulations for example.

- – Combining an antidiffusive advection scheme, the use of real vertical wind and the largest number of levels system-
5 atically permit to reduce plume diffusion. Qualitatively, the impact of the antidiffusive transport scheme on reducing vertical diffusion seems to be more pronounced than that of using real vertical winds instead of reconstructed ones: in Figure 7: plumes on the third row of Figure 7, with the Després and Lagoutière (1999) scheme and reconstructed vertical winds, are systematically less diffused than their counterpart on the second row with the Van Leer (1977) scheme and realistic vertical winds.

– Examination of the 4 simulations with 99 vertical levels shows that the Després and Lagoutière (1999) preserves much higher maximal concentrations in the plume: at the last simulation step, both simulations with the DL99 scheme exhibit maximum SO$_2$ concentrations in the 75-250 ppb range, while both simulations with Van Leer (1977) have maximal concentrations in the 10-25 ppb range little more than 48 hours after the eruption, and all other things being equal, choosing the Després and Lagoutière (1999) anti-diffusive transport scheme permits the simulated concentration to be at
least 4 times as strong as with the classical Van Leer (1977) scheme.

### 3.6 Parameters impact on SO$_2$ dispersion

In the aim to evaluate the SO$_2$ overall diffusion (vertical and horizontal) following the present Etna eruption, we have compared the minimum volume in which 50 % of SO$_2$ mass can be found for each time step and for each simulations. Volume evolution have been represented on Figure 8a. 20 levels simulations present the highest volume occupied, and 99 levels simulations the lowest volume occupied. The simulation which restricts most the diffusion is WRFW-DL-99 one, and by opposition, the one which contains less efficiently the plume is NODIV-VL-20.

To assess the impact of the applied vertical scheme or vertical wind strategy, the volume ratio evolution between VL and DL vertical scheme for each vertical levels number has been calculated - with the wind strategy fixed (NODIV). Likewise, the volume ratio evolution between NODIV and WRFW wind strategy for each vertical levels number has been calculated - with the vertical scheme fixed (VL). Volume ratio evolution have been summarized on Figure 8b. It appears that the DL vertical transport scheme strongly reduces diffusion compared to VL, as the volume ratios $Vol_{NODIV-VL}/Vol_{NODIV-DL}$ increase with time for the three level numbers configuration. Results are less clear for the vertical wind strategy, as the volume ratios $Vol_{NODIV-VL}/Vol_{WRFW-VL}$ slightly increase in most of the cases, with a stronger effect in the 20 vertical levels case. Finally, we observe that the combination of WRFW and DL parametrisation has consequently reduced the atmospheric diffusion, so that plume volume for WRFW-DL-20 is quite similar to the NODIV-VL-99 case: using a realistic vertical wind field and an antidiffusive scheme is, for this criterion, as efficient as refining the model vertical resolution by a factor of 5.

As discussed in Zhuang et al. (2018); Eastham and Jacob (2017), reducing vertical diffusion has a direct impact on reducing horizontal diffusion as well. Figure S6 shows integrated SO$_2$ colums for six model configurations 48h after the eruption. We can observe here that, in terms of maximum value of integrated SO$_2$ column, WRFW-DL-50 produces a maximum value slightly stronger than NODIV-VL-99, with strikingly similar horizontal structures. Also, in spite of the fact that 20 vertical layers are clearly not sufficient to reproduce the plume, we can see that, if we take the 99-level simulations as reference points, the output of the WRFW-DL-20 is clearly better than NODIV-VL-20 in terms of horizontal structure and maximal column values, which is visible in many aspects: orientation of the low-level plume from south-eastern Italy to the Aegean sea and north-south from the Aegean sea to eastern Libya, stronger maximal values of SO$_2$ columns in the main part of the plume above the Middle-East. This qualitative comparison is in line with the results of Zhuang et al. (2018); Eastham and Jacob (2017) in the fact that improving vertical resolution will substantially reduce the horizontal spread in simulated plumes as well. We have also calculated the minimum area containing more than 50% of the SO$_2$ mass (Figure S7), showing that the WRFW-DL simulations concentrate 50% of the plume mass in an area at least twice as small as their NODIV-VL counterparts.

### 3.7 Evaluation of SO$_2$ dispersion with similar vertical extension at injection

To evaluate the impact schemes and vertical resolution would have with a similar vertical extension at injection, new simulations have been conducted imposing an identical vertical distribution at the first time (spreading vertically the emited mass over the same thickness in the 50 and 99-level simulations than it has in the 20-level simulation). Simulations have been conducted for 20, 50 and 99 vertical levels, for WRFW-DL and NODIV-VL parameters, representing a total of six simulations. Results

have been displayed in supplements, on Figure S5. It can be seen on Figure S5 (left) that all plumes have the same initial volume regardless of vertical resolution, which was not the case in the previous case (c.f. Figure 8a). With a larger vertical extension of the plume at injection, volumes are higher than in the "unique cell injection" cases, but resolution and transport scheme influence in the same way the evolution of plume (considering its volume). Figure S5 (right) shows evolutions of $SO_2$ highest column vertical profile, similar to Figure 7. This new set of experiments show that, even when getting rid of the initial distorsion due to sharper injection profiles in the simulations with the most refined vertical distributions, the increase in plume volume is much slower in the 99-level simulations than in the 20-level simulations. The final volume is about 4 times smaller in the 99-level simulations compared to their 20-level counterparts. A similar factor in volume reduction is obtained by changing strategy from VL-NODIV to DL-REALW. In total, final plume volume in the worst-case NODIV-VL-20 simulation is about 20 times bigger than final plume volume in the best-case WRFW-DL-99 simulation. Figure S5 (right panel) shows that simulation WRFW-DL-99 is able to reproduce plume thinning under the effect of wind shear, with the plume getting thinner at the end the simulation than it was at the beginning.

## 4   Conclusion

The Etna eruption of March 18, 2012 has been modelled using the CHIMERE chemistry-transport model in the aim to propose and test strategies to improve representation of atmospheric vertical diffusion which has previously been described in multiple studies (Colette et al., 2011; Boichu et al., 2015; Mailler et al., 2017, e.g.) as over-diffusing, inducing an excessive spread of the simulated plumes. First, the sensitivity to plume injection height and profile have been evaluated, following the plume trajectory with satellites retrievals. It appeared in these tests that the trajectory is highly sensitive to the injection altitude. The intermediate option (injection at 12 km) has been retained and tests and comparisons have been made with this injection altitude. No significant impact of plume injection profile has been observed and the most simple case of a unique altitude emission has been retained.

In order to reduce the excessive spread of the plume in the vertical direction due to numerical diffusion, three possible approaches have been tested: increasing the number of levels (20, 50 and 99 level simulations have been performed), using the anti-diffusive scheme of Després and Lagoutière (1999) instead of the classical Van Leer (1977) second-order slope-limited scheme, and using realistic vertical winds instead of vertically reconstructed vertical winds, as it is usually done in CTMs (Emery et al., 2011), to the expense of tracer mass conservation. Our results show that, as expected and as already shown in earlier studies, 20 vertical levels are clearly not sufficient to usefully represent any property of vertical transport and dispersion of this plume, and that increasing the number of vertical layers to 50 or to 99 brings significant added value in all respects: horizontal trajectories are improved compared to satellite measurements, vertical diffusion is reduced and maximal concentrations are preserved better. Also very effective is the use of the Després and Lagoutière (1999) anti-diffusive transport scheme instead of the Van Leer (1977) scheme. To our knowledge, this scheme has never been used in chemistry-transport studies, and we show here that this strategy has a very strong potential in preventing simulations to be affected by excessive vertical diffusion without dramatically increasing the number of vertical levels. In our simulations, using the Després and

Lagoutière (1999) scheme with 50 levels only has lead to performances that are comparable to the ones obtained with the Van Leer (1977) scheme and 99 levels (compare Figure 7c to Figure 7h or Figure 7f to Figure 7k). With an equivalent number of vertical levels, maximum concentrations in the plume after slightly more that 48 hours of atmospheric transport are about four times as strong in a simulation with the Després and Lagoutière (1999) scheme than in the same simulation but with the Van Leer (1977) scheme. In addition, increasing vertical resolution might give a false appearance of accuracy to the result when plume injection altitude is not known with a good precision. Finally, it has been shown than using realistic vertical winds instead of reconstructed vertical winds also improve the horizontal trajectories of the plume, when compared to satellite observations, and reduce plume diffusion in terms of minimum volume containing at least half of the plume mass. It needs to be recalled here that this strategy does not guarantee mass tracer conservation. Even though its impact in this respect has been shown to be quite minor in our study, except in the simulations with 20 vertical levels where it generated an initial excess in tracer mass of about 15%, this characteristic needs to be kept in mind, accounted for and monitored by potential users of this strategy.

These different strategies need to be further studied in different cases to determine whether they can be generalized in CTMs in order to reduce vertical diffusion issues for all pollutants and all kinds of problems, if they are useful only for long-distance transport of inert plumes as we simulated here. For example, how does the Després and Lagoutière (1999) perform in preserving smooth gradients, transitions between the PBL and the free troposphere, or gradients in ozone concentrations at the tropopause ? Regarding the use of realistic vertical wind fluxes interpolated from meteorological outputs, does it help reduce excessive ozone transport through the tropopause as identified by Emery et al. (2011) ? Can this method be generalized to the general chemistry-transport modelling of the troposphere in spite of the mass conservation issues that are intrinsic to this method ?

This study is a call to reopen the issue of limiting vertical mass diffusion in eulerian CTMs: complementary to Zhuang et al. (2018) who emphasized on the need for sufficient vertical resolution, which is confirmed by the present study, we propose two new approaches solve this long-standing problem, including anti-diffusive transport schemes and better representation of vertical mass fluxes throughout the troposphere. Our results show that these new approaches on the vertical direction also permit to reduce horizontal diffusion and that this reduction can be achieved not only by increasing vertical resolution, as shown by these authors, but also by using the Després and Lagoutière (1999) transport scheme as an alternative to classical schemes.

**Code and data availability**

The source code for the CHIMERE model (Mailler et al., 2017) is available on: https://www.lmd.polytechnique.fr/chimere/. WRF source code is available on: https://github.com/wrf-model/WRF/. OMI Data (Levelt et al., 2006; McCormick et al., 2013) are available on the NASA GIOVANNI platform: https://giovanni.gsfc.nasa.gov/giovanni/. IASI data (Carboni et al., 2012, 2016) are available contacting the authors. $SO_2$ (Salerno et al., 2018) flux measurement data are available contacting the authors. Simulation outputs are available contacting the authors.

**Authors contributions**

S.M. designed the experiments and carried them out. H.G., E.C. and P.S. were responsible for the processing of IASI instrument $SO_2$ retrieval dataset. G.S., T.C. and S.M prepared eruption emission data. S.M. adapted the model code and performed the simulations. M.L carried out the instruments-model and model-model comparisons. M.L. and S.M. prepared the manuscript
5    and all authors contributed to the text, interpretation of the results and reviewed the manuscript.

*Acknowledgements.*  This study has been supported by AID (Agence de l'Innovation de Défense) under grant TROMPET. Simulations have been performed on the Irene supercomputer in the framework of GENCI GEN10274 project. This work has been supported by the Programme National de Télédétection Spatiale (PNTS, http://www.insu.cnrs.fr/pnts ), grant n°PNTS-2019-9.

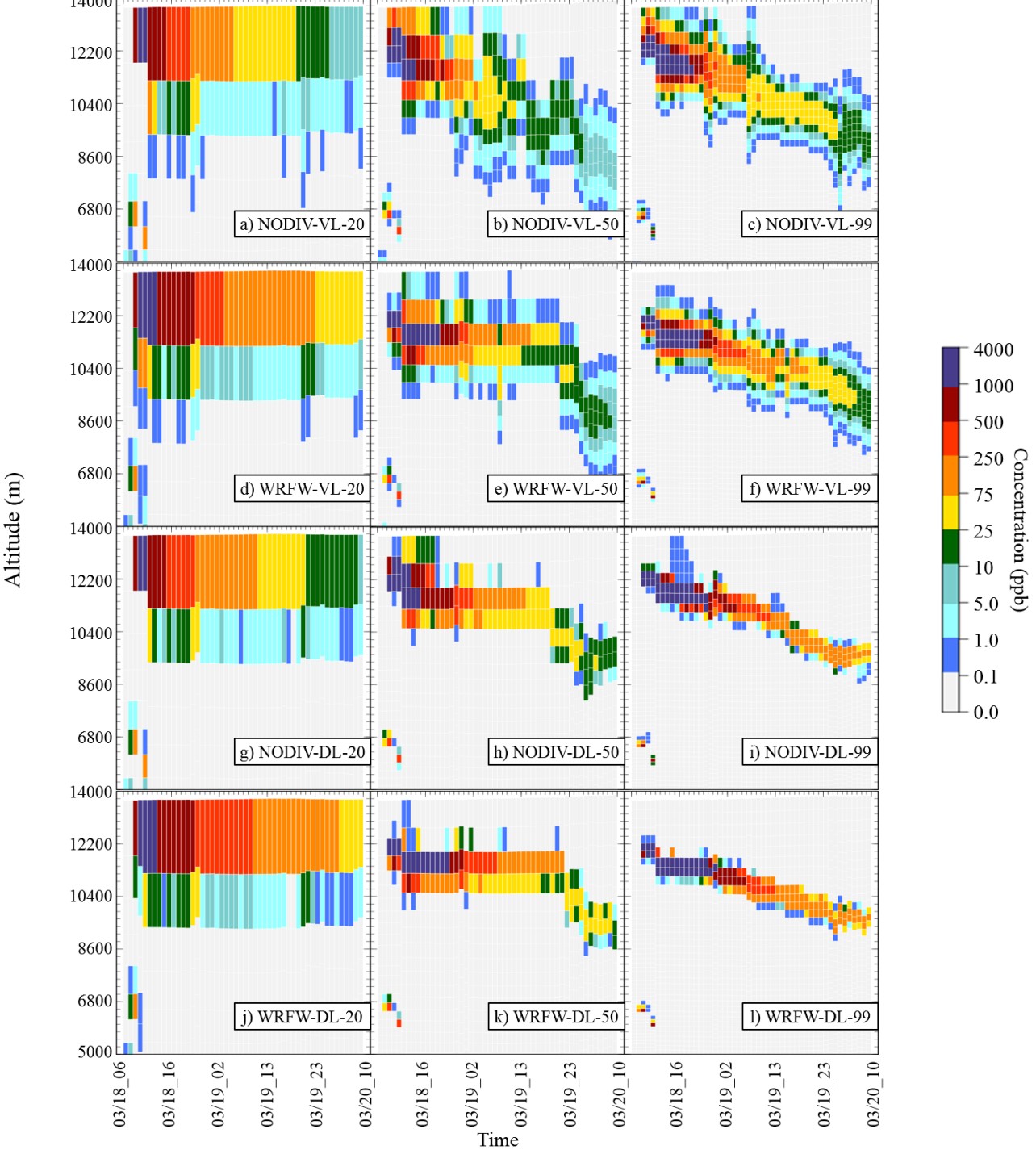

**Figure 7.** Evolution of SO$_2$ vertical profile (in ppb) corresponding to the maximum column for each step after the Etna eruption, for each tested model configurations. $1^{st}$ row: NODIV-VL; $2^{nd}$ row: NODIV-DL; $3^{rd}$ row: WRFW-VL; $4^{th}$ row: WRFW-DL. Left: 20 vertical levels; Center: 50 vertical levels; Right: 99 vertical levels. WRFW simulations values have been corrected to fit NODIV strategy masses.

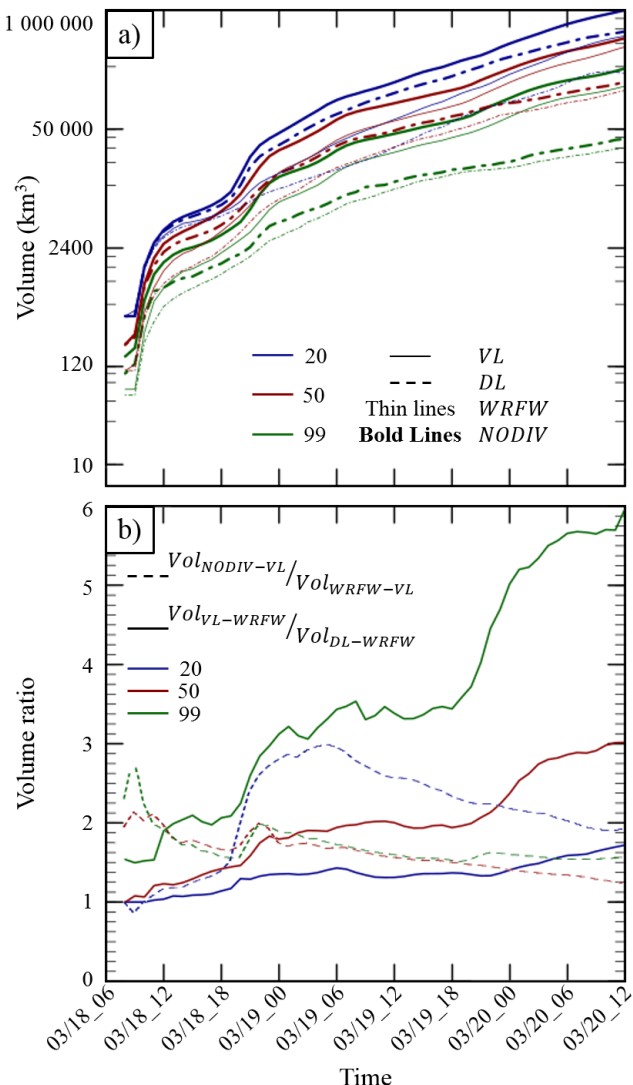

**Figure 8.** a) Minimum volume evolution calculated for 50 % of SO$_2$ total mass in the atmosphere. b) Volume ratio evolution by parameters.

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
