# Peer review of "New strategies for vertical transport in chemistry-transport models: application to the case of the Mount Etna eruption on March 18, 2012 with CHIMERE v2017r4"

_Geoscientific Model Development, 2020_

## Referee Comment (RC1) · Anonymous Referee #1 · 16 May 2020

The authors identify three possible options to try and address the issue of excessive vertical diffusion in global, Eulerian, atmospheric models. They test three different solutions: the use of an antidiffusive numerical scheme for vertical transport; increasing the number of vertical layers; and using an alternative scheme to calculate vertical mass fluxes. They find that all three approaches reduce the degree of vertical diffusion, although to different extents. The conclusion drawn is that, while these approaches do not solve the problem of vertical diffusion, some combination of the approach might help to solve this problem.

[Figure]

I agree fully with the authors that this question is timely, underrepresented in the current literature, and important. The approaches they use are innovative and appropriate, although I have some concerns regarding the results and presentation. With one caveat it appears that the data also generally support the conclusion, which is an incremental but important advance of the conversation surrounding the accuracy of chemistry transport models.

With this in mind, I believe that this paper is also ideally suited for the audience of Geoscientific Model Development. With some revisions, I also believe that it is appropriate for publication. However, there are some concerns I would like to see addressed first.

Major concerns

Firstly, this paper seeks to address two major concerns regarding vertical transport: 1. Vertical transport is poorly represented in most modern chemistry transport modeling efforts, resulting in excessive numerical (and eventually horizontal) diffusion; and 2. The naïve, or brute-force, solution to this – increasing the number of levels in the simulation – is expensive. This paper has done an excellent job of exploring answers to the first question, but does not provide any insight into the second. The two "smart" solutions which the authors propose have their own downsides; the Després and Lagoutière (hereafter DL) advection scheme, while antidiffusive, is also only first-order accurate, while the "directly interpolated winds" (hereafter WRFW) approach violates mass conservation. The utility of the paper would be significantly increased if the authors gave a quantitative assessment of the computational overhead associated with each method and compared it to that associated with the naïve approach. Timing alone, in terms of the number of CPU-hours spent on each simulation, would help with this.

Similarly, the lack of mass conservation in the WRFW approach causes serious concern. I applaud the authors for their frankness in discussing this limitation. However I believe that a full understanding of the advantages and drawbacks of each approach

demands a fuller discussion of this issue than is currently given in Section 3.2. In Figure 3, it is not clear to the reader why the total domain mass differs so much between each simulation, and it is critically important to the core question of the paper to know why the mass is changing. Specifically, it would help greatly if the authors could quantify on or with Figure 3: 1. How much mass has been (erroneously) lost through the domain upper boundary, based on integrated vertical mass fluxes at the upper boundary; and 2. How much mass has been lost through the domain side boundaries, based on integrated horizontal mass fluxes at the domain boundary. These quantities should enable the authors (and reader) to determine how much of the mass at a given time is spurious, and the degree to which loss through the boundaries is offsetting artificial mass production. On this note, on lines 2-3 of page 14, the authors mention that the "spurious evolutions in tracer mass become weaker, less than 5%" once the plume is more diffuse. Does this really mean "the total domain mass is <105% of the total emitted mass", or is it saying that the amount of mass created spuriously in each time step is <5% of the current domain total? I assume the former, but if so, does this really mean that the error is <5%, or just that the additional spurious mass is now offset by some loss of mass through the domain boundaries?

A broader concern which does not appear to be discussed in detail is the fact that the simulation is driven by fields which are sometimes at a lower vertical resolution. CHIMERE is driven by WRF, running with 33 models, but CHIMERE interpolates this data to its target vertical resolution (Briant et al 2017). Is this interpolation done in a divergence-conserving fashion? If not, does this constitute an uncontrolled-for additional term, in the sense that different vertical grids could introduce different amounts of artificial divergence?

Finally, the authors rely heavily on the trajectory of the plume as a metric of the simulation's fidelity. While the equation to determine error (equation 16) is an interesting formulation, it would be helpful to provide a more quantitative assessment of the amount of numerical diffusion. Variation in the maximum volumetric mixing ratio, the

total area of the plume above some minimum VMR, or the total entropy would be useful for quantifying how much numerical diffusion is being introduced. This would also allow the authors to account for the effect that spurious vertical diffusion can have in accelerating spurious horizontal diffusion (relevant papers discussing this issue and metrics of numerical diffusion are e.g. Rastigejev et al 2010, Lauritzen and Thuburn 2012, Eastham et al 2017, Zhuang et al 2018).

Minor comments

I believe that there is an error in equation 15. Using the case of a local maximum (i.e. the first term of the Min operator is negative or zero), the estimated cell boundary VMR ends up being the cell mean VMR + 1, when it should presumably by the cell VMR only (specifically if this is meant to recreate the Godunov donor cell scheme for that condition). Although only a technical error, this is critically important to verification of the rest of the paper.

Section 2.1: it would be helpful to have details on how the vertical layers are placed (i.e. more detail on the different grid discretizations), and where the cell edges lie relative to the WRF vertical grid.

P12 L6: 'independant' should be 'independent'

P18 L21: Currently this line appears to compare the Després and Lagoutière scheme to itself. Should the second instance actually be "van Leer (1977)"?

P20 L2: Why is increasing vertical resolution only meaningful in cases where plume injection altitude is known? I feel that this statement needs to be better qualified. A reduction in numerical diffusion should always correspond to an improvement in simulation fidelity, even if the initial conditions include error.

Finally, the paper has some minor grammatical errors throughout (e.g. page 1 line 15, "The CHIMERE CTM has previously been used to assess Eyjajallajökull eruption possible impact on air quality" should be "..to assess the possible impact of the eruption of

Eyjajallajökull on air quality"). I hesitate to bring these up as the errors are almost always very minor and do not impact the science of the paper, and it is usually possible to determine the authors' intended meaning. However, these issues do compromise the readability, and as such I would recommend the authors take another sweep through the paper to correct such issues.

References

Briant, R., Tuccella, P., Deroubaix, A., Khvorostyanov, D., Menut, L., Mailler, S. and Turquety, S.: Aerosol–radiation interaction modelling using online coupling between the WRF 3.7.1 meteorological model and the CHIMERE 2016 chemistry-transport model, through the OASIS3-MCT coupler, Geoscientific Model Development, 10(2), 927–944, 2017.

Eastham, S. D. and Jacob, D. J.: Limits on the ability of global Eulerian models to resolve intercontinental transport of chemical plumes, Atmos. Chem. Phys., 17, 2543–2553, 2017.

Lauritzen, P. H. and Thuburn, J.: Evaluating advection/transport schemes using interrelated tracers, scatter plots and numerical mixing diagnostics, Q.J.R. Meteorol. Soc., 138(665), 906–918, 2012.

Rastigejev, Y., Park, R., Brenner, M. P. and Jacob, D. J.: Resolving intercontinental pollution plumes in global models of atmospheric transport, J. Geophys. Res., 115(D2), D02302, 2010.

Zhuang, J., Jacob, D. J. and Eastham, S. D.: The importance of vertical resolution in the free troposphere for modeling intercontinental plumes, Atmos. Chem. Phys., doi:10.5194/acp-2017-1124, 2018.
* * *

---

## Referee Comment (RC2) · Anonymous Referee #2 · 20 May 2020

I - General comments

This manuscript presents new numerical modeling approaches to represent vertical transport of pollutants plumes in the upper troposphere with Eulerian Chemistry-Transport Models (CTM). The aim is to limit the excessive vertical diffusion of the plumes of pollutant in this kind of numerical representation.

Different numerical strategies are considered to address this issue : different vertical wind diagnosis, different advection scheme and different vertical resolution. The sensitivity of the simulation of a plume transport event to the different numerical choices considered in this work is evaluated on the case of the Mount Etna's eruption of March 18, 2012.

Its a topic of scientific interest and certainly within the scope of Geoscientific Model Development. The general presentation of the work is logically and clearly organized. However the added value of this work could be improved with the clarification and/or the development of some results.

II - Specific Comments

Section 2.1

p4 l7 and l13: Could the authors precise which CHIMERE version has been used? 2013, 2017 or 2016?

p4 l14 : The horizontal resolution of the WRF simulation should be mentioned.

p4 l17 : The authors should provide the limits of the vertical layers (at least in supplement with figure S4).

p4 l16-17 : There is an in-depth discussion on the relationship between horizontal resolution and vertical resolution in the article by Zhuang et al. (2018) that the authors cite in the introduction, but nothing is said here on this subject. Beyond this point, it is weird to see that the chosen vertical extension of the domain does not provide any possibility to reproduce the highest part of the plume as seen by the observations (cf. figure 2e, S3 and S4). All the more so when we see that the meteorological simulation would allow the domain to be extended. Could the authors explain how they chose the different vertical resolutions tested?

p4 l17 : These different vertical resolutions rely on an oversampling of the same simulated meteorological fields. Could we expect to get significantly different vertical profile with a meteorological simulation carried out with a finer vertical grid?

p4 l17 : Only part of the 33 vertical levels of the meteorological grid is used for the interpolation on the dispersion grid. The number of levels concerned could be specified in this paragraph.

p10 l11 and p12 l5: The comparison between the different vertical resolutions involve an aspect which may deserve a bit more detailed discussion. Which kind of boundary conditions are applied for the pollutant concentrations? With a plume injection in the last layer of the model (at least when 20 levels are used) it seems that the boundary conditions could play a role. What happens when the flux is downward oriented? (here again the choice of a larger vertical extension would be more relevant).

p13 l1-4 : Could the authors provide the levels concerned in these tests?

p13 l5-7 : The authors mention an "injection to a unique altitude". It implies the different simulations with the different vertical resolution do not start with the same vertical extension of the plume. It would be interesting to isolate the impact of this initial discrepancy that cannot be associated to an excessive diffusion of the advection scheme. I guess this could be done with a 50 or 99 levels simulations run with an injection uniformly distributed over the different layers corresponding to the injection layer of the 20 levels simulation.

p14 section 3.3 : With the location of "the model column with the strongest vertically integrated SO2 content" the authors have chosen a very aggregated indicator for the comparison between satellite soundings and model results. I assume this choice was made for sake of simplicity in the presentation of the results. However, seeing that the configuration option can lead to some plume splitting, it would be interesting to have more information concerning the horizontal extension of the plume in the different cases.

p14 l23 : It is not clear to me if the results in figure 5 present average over different configuration options. For instance in the first panel, the simulations with the different advection scheme are compared. Do the number are averages over the different

vertical resolutions (the vertical resolution is not mentioned either in the text or in the label of the figure)? Does this imply that there is few interaction between the tested options? From figure 4 we can see that the WRFW-DL-99 simulation is not the closest to the observation at the final stage. This may not be the intuition get from the results presented.

p16 l18-20 : Could the authors precise how the distribution are built? It is not clear for me if it represents different time steps, different locations, or a mix? Are the observations uncertainties are represented in this figure?

p17 l6-8 : These lines are frustrating from my point of view. The authors focus their work on the excessive vertical diffusion in the dispersion model and the only comparison of the model results to observation concerning the plume vertical extension indicates that this plume property is underestimated. Could the authors provide a more in-depth discussion concerning this point? Some considerations concerning the time evolution of the maximum concentration (modeled and observed) could be useful here to convince the readers that a less diffusive treatment of the advection is really suitable. Since the transport in this application is linear, even a normalized comparison to the "initial" ($\sim$ sounding number 1) maximum concentration would be useful.

III - Technical corrections

p5 l11 : It seems an "overbar" is missing for notation consistency.

p11 label table 3 : The last sentence should probably be in the label of Figure 2.

p18 l21 : erroneous citation

---

## Author Comment (AC1) · 11 Jul 2020

**1 Answer to Anonymous referee #1, received 16 may 2020**

We wish to thank the referee for his/her helpful comments. The comments of the referee are in bold, our answers in normal black, the new elements added to the text are in blue.

**1.1 Major comments**

**Firstly, this paper seeks to address two major concerns regarding vertical transport: 1. Vertical transport is poorly represented in most modern chemistry transport modeling efforts, resulting in excessive numerical (and eventually horizontal) diffusion; and 2. The naïve, or brute-force, solution to this increasing the number of levels in the simulation is expensive. This paper has done an excellent job of exploring answers to the first question, but does not provide any insight into the second.**

**The two "smart" solutions which the authors propose have their own downsides; the Després and Lagoutière (hereafter DL) advection scheme, while antidiffusive, is also only first-order accurate, while the "directly interpolated winds" (hereafter WRFW) approach violates mass conservation. The utility of the paper would be significantly increased if the authors gave a quantitative assessment of the computational overhead associated with each method and compared it to that associated with the naïve approach. Timing alone, in terms of the number of CPU-hours spent on each simulation, would help with this.**

| Parameters / Resolution | 20 | 50 | 99 |
|---|---|---|---|
| NODIV-VL | 943 | 1177 | 1376 |
| WRFW-VL | 938 | 1193 | 1380 |
| NODIV-DL | 957 | 1193 | 1389 |
| WRFW-DL | 936 | 1187 | 1302 |

Table 1: Number of CPU hours for each simulation setup

The number of CPU hours spent on each simulation is provided in Table 1 above. They do not fit any theoretical scaling. The scaling of the computational load relative to the number of vertical levels $n$ is known to be at least

proportional to $n$ (and proportional to $n^2$ if the CFL in the vertical direction constrains the timestep). Here our observed the scaling is sub-linear which is unexpected.

Here the configuration was 384 CPUs for CHIMERE and 128 CPUs for WRF. The configuration of CHIMERE is extremely light, with only 1 advected species and no chemistry, so that most likely the meteorological simulation, an extremely complex process with several prognostic variables, was using most of the CPU time, with the CHIMERE CPUs likely spending part of the time waiting for the input meteorological fields, at least in the lightest configuration with 20 model levels. It would have been more efficient in terms of computational time to use fewer CPUs for CHIMERE at least in the simulations with 20 levels to balance the load between meteorology and chemistry, but since the point here was to compare the results of the various simulations we preferred to choose an "all other things being equal approach" where the only change in configutaion between a simulation with 20 levels and its 99-levels counterpart is the number of levels.

This underloading of CHIMERE CPUs is very specific to the present configuration since we advect only one species (typically hundreds of species in a CTM simulation). We have observed that in full-fledge CHIMERE simulations with realistic chemistry and using pre-calculated meteorological fields the scaling of computational time according to the number of vertical levels is linear or superlinear.

Due to these limitations, we are unfortunately not able to use our results to provide a more precise information on computational cost.

**Similarly, the lack of mass conservation in the WRFW approach causes serious concern. I applaud the authors for their frankness in discussing this limitation. However I believe that a full understanding of the advantages and drawbacks of each approach demands a fuller discussion of this issue than is currently given in Section 3.2.**

**In Figure 3, it is not clear to the reader why the total domain mass differs so much between each simulation, and it is critically important to the core question of the paper to know why the mass is changing. Specifically, it would help greatly if the authors could quantify on or with Figure 3: 1. How much mass has been (erroneously) lost through the domain upper boundary, based on integrated vertical mass fluxes at the upper boundary;**

In Figure 3 (reproduced below), considering the "NODIV" simulations which are mass conservative, $SO_2$ mass loss is only due to fluxes through the model upper boundary. For this wind strategy, the differences between 20, 50 and 99 vertical levels are explained by the plume proximity to model upper boundary, which can be observed in Figure 7.

[Figure]

**Figure 3.** SO$_2$ mass evolution in model domain (kilotons). Line color indicates the vertical levels configuration, thickness indicates the vertical wind strategy considered. Dotted line represents the cumulated SO$_2$ mass emitted during Etna volcanic Eruption.

and 2. How much mass has been lost through the domain side boundaries, based on integrated horizontal mass fluxes at the domain boundary. These quantities should enable the authors (and reader) to determine how much of the mass at a given time is spurious, and the degree to which loss through the boundaries is offsetting artificial mass production.

On this note, on lines 2-3 of page 14, the authors mention that the "spurious evolutions in tracer mass become weaker, less than 5 %" once the plume is more diffuse. Does this really mean "the total domain mass is $< 5\%$ of the total emitted mass", or is it saying that the amount of mass created spuriously in each time step is $< 5\%$ of the current domain total? I assume the former, but if so, does this really mean that the error is $< 5\%$, or just that the additional spurious mass is now offset by some loss of mass through the domain boundaries?

The negative trend due to leakage through top of domain is observed mostly in the simulations with 20 and 50 levels. For the simulation with reconstructed wind, this leakage is the only term of mass loss: therefore, we can identify easily the magnitude of this term without additional calculation.

The idea of error compensation is interesting. However, a close look at the curves shows that the decreasing trend due to mass leak at top of model is present in the simulations with interpolated wind as well (thin lines in Fig. 3 reproduced above), and with a comparable magnitude. We think that the effect of mass balance inconsistency due to the divergence of wind field are visible in the small deviations of the curves corresponding to the WRFW simulations around this long-term trend, giving them a more wiggly aspect than those with NODIV which display only a slow and steady decrease. These small movements

$$\frac{\partial \widetilde{C}_{i,j,k}}{\partial t} + \left( \widetilde{F}_{i,j,k+\frac{1}{2}} - \widetilde{F}_{i,j,k-\frac{1}{2}} \right) + \left( \widetilde{F}_{i+\frac{1}{2},j,k} - \widetilde{F}_{i-\frac{1}{2},j,k} \right) + \left( \widetilde{F}_{i,j+\frac{1}{2},k} - \widetilde{F}_{i,j-\frac{1}{2},k} \right) = -\varepsilon_{i,j,k}, \tag{6}$$

are indifferently positive or negative. We think that the effect of this term is not necessarily the effect of "additional spurious mass", but can be indifferently positive and negative as shown by Eq. 6. Actually, errors in discretized calculation of divergence will tend to compensate each other between neighbouring cells, so that we think that the relatively weak effect of the mass inconsistency term as soon as the plume is spread over many cells is due to this error compensation, between neighbouring cells:

If, for example, $\widetilde{F}_{i+\frac{1}{2},j,k}$ is overestimated, this will introduce a negative contribution in $\varepsilon_{i,j,k}$ but a positive and opposite contribution on $\varepsilon_{i+1,j,k}$.

This also explains why the $\varepsilon_{i,j,k}$ term has a much more drastic impact in the first hours of the eruption, because in these hours a substantial part of the total tracer mass is concentrated in one single cell above the vent: then the sign and magnitude of the error term $\varepsilon_{i,j,k}$ in this precise cell becomes critically important and no error compensation occur since the opposite errors on neighbouring cells will act on much smaller tracer concentrations.

Two new paragraphs have been added in Section 3.2 to discuss these points :

In the simulations with the reconstituted non-divergent wind field, substantial mass leak through the top of model can be observed as soon as the injection starts in the 20-level simulation (in which injection is done in the highest model level): the mass of tracer present in the domain never exceeds 85% of the expected mass. For the simulation with 50 vertical levels, this phenomenon is also visible. Another strong episode of mass leak through model top occurs in the simulations with 20 and 50 vertical levels and with reconstructed wind fields from March 18, 18UTC to March 19, 00UTC. This episodes causes an additional drop in tracer mass of  20% in the simulation with 20 levels, 5% in the simulation with 50 vertical levels. This episode of leak also affects the simulation with 20 vertical levels and with interpolated wind fields, reducing tracer mass concentration by about 10% from March 18, 18UTC to March 19, 00UTC. In these three simulations (20 and 50 levels with non-divergent winds, 20 levels with interpolated winds), a continuous decreasing trend in tracer mass is observed throughout the simulation. This drop is directly attributable to leak through model top since the tracer plume is far away from the horizontal boundaries of the domain.

And:

No physical process can explain this overshoot, and it is directly attributable to the choice of lifting the mass conservation constraint in the formulation of transport in order to permit the use of a realistic wind field. If we take March 19, 00UTC as a reference time at which the eruption is terminated, the first strong event of leak through model top is terminated as well, we can observe that the mass evolution in all three WRFW simulations undergoes small variations from

one hour to the next but stay confined in very narrow ranges : 3.3 to 3 kt for the simulation with 20 vertical levels, with a decreasing trend attributable to leakage through model top, 3.1 to 3.25 for the simulation with 50 levels and 2.9 to 3.1 kt for the simulation with 99 vertical levels. The fact that these variations in total mass become marginal in this latter part of plume advection, when the plume is spread over a large geographic areas reflect the fact that numerical errors in the evaluation of divergence mechanically tend to compensate each other between neighbouring cells so that their global impact on a plume that is dispersed over many cells is small.

**A broader concern which does not appear to be discussed in detail is the fact that the simulation is driven by fields which are sometimes at a lower vertical resolution. CHIMERE is driven by WRF, running with 33 models, but CHIMERE interpolates this data to its target vertical resolution (Briant et al 2017). Is this interpolation done in a divergence-conserving fashion? If not, does this constitute an uncontrolled-for additional term, in the sense that different vertical grids could introduce different amounts of artificial divergence?**

The interpolation of the wind fields is done in a linear fashion which in principle is divergence conserving, but CHIMERE interpolation works directly on winds and not mass fluxes which actually may bring some additional errors in divergence. Our concern was to have all simulations forced with the exact same meteorological simulation, and we decided to retain the typical number of levels that is used in CHIMERE (Briant et al., 2017). The statement that "different vertical grids could introduce different amounts of artificial divergence" is therefore correct. We explicitly draw the reader's attention towards this point in section 2.2.1 of the revised version:

$\varepsilon_{i,j,k}$ depends on the resolution of the meteorological model (which is identical for all our simulations), and on the resolution of the chemistry-transport model, so that this error term that essentially traduces divergence errors due to interpolation depends on the vertical resolution of the model. It is identical between simulations that have the exact same number of domains. Choosing interpolation strategies that reduce this error term is a promising path to mitigating excessive vertical diffusion, as discussed in Emery et al. 2011, but is not investigated here.

**Finally, the authors rely heavily on the trajectory of the plume as a metric of the simulation's fidelity. While the equation to determine error (equation 16) is an interesting formulation,**

This is true because the plume's horizontal location is the only reliable observation that we had, due to the large uncertainty and error bars in the satellite retrievals of its altitude. Therefore, indicators like the one in Eq. 16 were, unfortunately, our only way to provide a comparison of model simulations with real-world data. We agree that this measure is only an indirect way to observe potential improvements in the vertical direction and reduction in plume diffusion.

**(...)it would be helpful to provide a more quantitative assessment of the amount of numerical diffusion. Variation in the maximum**

**volumetric mixing ratio, (...)**

Figure 7 (reproduced below) displays the highest column vertical profile evolution for each simulation. It can be observed that $SO_2$ mixing ratio is highly impacted by diffusion parameters chosen (please note that the scale use is irregular), and that simulations with the WRFW-DL configurations preserve a much higher maximal VMR than their conterparts with NODIV-VL.

Also, we believe that Figure 8 as well as Figures S5 and S7 in the supplements that have been added in the revised version bring additional elements in this line. Generally speaking, we have chosen to look at a more synthetic parameter like the minimal volume containing 50% of mass plume rather than a value of maximal VMR, which is more dependant on the details of all simulations. Figures 8a and S5 can be directly interpreted in terms of VMR, since the typical VMR in the plume is inversely proportional to plume volume.

**(...) the total area of the plume above some minimum VMR, or the total entropy would be useful for quantifying how much numerical diffusion is being introduced.**

A calculation very similar to the one suggested by the Referee on area is already present in the manuscript (Section 3.6, Fig 8 of the submitted manuscript and Fig. S5 of the revised manuscript). Here we propose to use the minimum volume containing at least half of the $SO_2$ mass as a synthetic indicator of how much the plume has been diffused. This is very similar to the proposal of calculating the area above some minimum VMR except that we chose to did it in 3d with volumes instead of areas, and we thought that calculating the volume containing at least half of the plume was a useful method to avoid introducing an arbitry threshold on VMR.

On Figure 8b), we calculate the volume ratios for each parameters (i.e. WRFW vs NODIV; DL vs VL) to provide a quantitative assessment of diffusion reduction on 3 dimensions. To illustrate the differences implied on for plume's surfaces, Figure S6 and Figure S7 (in suppl.) have been added to show the horizontal dispersion of plume on various simulations after 2 days.

We believe that entropy is delicate to interpret for many people including ourselves particularly when, as it is suggested here, we do not speak of a thermodynamic entropy of air but on the artificial construction of a mathematical entropy value for a tracer distribution. We agree that entropy of tracer concentration fields is a useful way of measuring numerical diffusion but we feel that discussing issues in terms of plume volume as we have done is much easier to interpret for the particular case we treat here, as we deal with a physical quantity whose absolute value has a meaning.

**This would also allow the authors to account for the effect that spurious vertical diffusion can have in accelerating spurious horizontal diffusion (relevant papers discussing this issue and metrics of numerical diffusion are e.g. Rastigejev et al 2010, Lauritzen and Thuburn 2012, Eastham et al 2017, Zhuang et al 2018).**

We agree with the Reviewer that more discussion on this point was useful. The results we obtain are in line with Eastham 2017 and Zhuang 2018: reduction of vertical diffusion has a direct impact on horizontal diffusion as well. Here in

[Figure]

**Figure 7.** Evolution of SO$_2$ vertical profile (in ppb) corresponding to the maximum column for each step after the Etna eruption, for each tested model configurations. $1^{st}$ row: NODIV-VL; $2^{nd}$ row: NODIV-DL; $3^{rd}$ row: WRFW-VL; $4^{th}$ row: WRFW-DL. Left: 20 vertical levels; Center: 50 vertical levels; Right: 99 vertical levels. WRFW simulations values have been corrected to fit NODIV strategy masses.

[Figure]

**Figure S6.** Volcanic plume integrated column dispersion on march 20$^{th}$ at 11 A.M. UTC (2 days after the eruption).

[Figure]

**Figure S7.** Minimum surface evolution calculated for 50 % of SO$_2$ total mass in the atmosphere.

the revised version we insist on the finding that this reduction can be obtained not only by improving resolutio but also, to some extent, by the approaches we advocate in the manuscript. A long discussion on this point has been introduced in section 3.6 based on new figures S6 and S7, and a corresponding statement is added in the conclusion as well.

**1.2 Minors comments**

**I believe that there is an error in equation 15. Using the case of a local maximum (i.e. the first term of the Min operator is negative or zero), the estimated cell boundary VMR ends up being the cell mean VMR + 1, when it should presumably by the cell VMR only (specifically if this is meant to recreate the Godunov donor cell scheme for that condition). Although only a technical error, this is critically important to verification of the rest of the paper.**

We are deeply grateful to the Referee for this in-depth investigation of our equation. This has permitted us to realize that there was actually a missing multiplicative factor in the equation and that this mistake would have made reproduction of our results in another model very difficult. The correct equation is as follows:

$$\bar{\alpha}_{s,k+\frac{1}{2}} = \alpha_{s,k} + \frac{1-\nu}{2} \text{Max}\left[0, \text{Min}\left(\frac{2}{\nu}\frac{\alpha_{s,k} - \alpha_{s,k-1}}{\alpha_{s,k+1} - \alpha_{s,k}}, \frac{2}{1-\nu}\right)\right] \times (\alpha_{s,k+1} - \alpha_{s,k}), \tag{1}$$

Even though the last factor was missing, the Referee's interpretation of the behaviour of Eq. 15 is correct and Eq. 15 would result into $\bar{\alpha}_{s,k+\frac{1}{2}} = \alpha_{s,k+1}$ in case of a local maximum, which would in our opinion lead to catastrophic instabilities since mass could never escape from a maximum whose downwind neighbour has zero VMR. As stated in the next sentence of the paper ("if $((\alpha_{s,k} - \alpha_{s,k-1})(\alpha_{s,k+1} - \alpha_{s,k}) \leq 0)$, no interpolation is performed and the scheme falls back to the simple Godunov donor-cell formulation"). This sentence may suggest that in case of a maximum the equation naturally falls back to the Godunov donor-cell formula. This is not the case. As we state more clearly in the revised version, Eq. 15 is applied if, and only if, the considered cell is not a local extremum, otherwise $\bar{\alpha}_{s,k+\frac{1}{2}} = \alpha_{s,k}$ is enforced:

As above, Eq. 15 is not applied in the case of a local extremum $((\alpha_{s,k} - \alpha_{s,k-1})(\alpha_{s,k+1} - \alpha_{s,k}) \leq 0)$. In this case, $\bar{\alpha}_{s,k+\frac{1}{2}} = \alpha_{s,k}$ is imposed and the scheme falls back to the simple Godunov donor-cell formulation

The same precision is broght for the Van Leer scheme (Eq. 14) since our initial formulation was suffering from the same ambiguity.

**Section 2.1: it would be helpful to have details on how the vertical layers are placed (i.e. more detail on the different grid discretizations), and where the cell edges lie relative to the WRF vertical grid.**

The various vertical resolutions can be compared on Figure S4 of the revised version.

[Figure]

**Figure S4.** Center : Maximum concentration altitude evolution (IASI and CHIMERE), IASI brackets indicate values' uncertainties and CHIMERE brackets indicates cell's bottom and top. Right : Model vertical levels distribution for the 3 configurations from surface to top.

A sentence has been added to the manuscript :

The WRF model has been run with 33 vertical levels from surface to 55 hPa (28 levels are into 1013-150 hPa range), and with an identical horizontal grid.

**P12 L6: 'independant' should be 'independent'**

Modification has been done.

**P18 L21: Currently this line appears to compare the Després and Lagoutière scheme to itself. Should the second instance actually be "van Leer (1977)"?**

Indeed, this has been modified.

**P20 L2: Why is increasing vertical resolution only meaningful in cases where plume injection altitude is known? I feel that this statement needs to be better qualified. A reduction in numerical diffusion should always correspond to an improvement in simulation fidelity, even if the initial conditions include error.**

We agree with the reviewer that this statement needs to be better qualified. However, we still believe that when increasing accuracy, the probability that the model vertical distribution is totally separated from the real vertical distribution increases. It is true however that, most likely, the qualitative features of the plume including its concentration may be better reproduced in this case even though possible at the wrong location. Therefore, we replaced the question statement by the following which we believe is more precise:

In addition, increasing vertical resolution might give a false appearance of accuracy to the result when plume injection altitude is not known with a good precision.

[Figure]

**Figure 1.** Satellite trajectory of the Etna volcanic plume (black line) built combining information from IASI and OMI instruments. CHIMERE simulated trajectory depending on $SO_2$ injection altitude of emissions (light and dark green lines - respectively NODIV-VL-99 and WRFW-DL-99). The grey area represents the CHIMERE simulation domain. White triangle indicates Mount Etna location.

**1.3  Minor grammatical errors**

**page 1 line 15, "The CHIMERE CTM has previously been used to assess Eyjajallajökull eruption possible impact on air quality" should be "..to assess the possible impact of the eruption of Eyjajallajökull on air quality").**

Phrase formulation has been modified.

**I hesitate to bring these up as the errors are almost always very minor and do not impact the science of the paper, and it is usually possible to determine the authors' intended meaning. However, these issues do compromise the readability, and as such I would recommend the authors take another sweep through the paper to correct such issues.**

We have performed a thorough checking of grammar and spelling in the manuscript and corrected these slips as best we could.

---

## Author Comment (AC2) · 11 Jul 2020

**1 Answer to Anonymous referee #2, received $20^{th}$ may 2020**

We wish to thank the referee for his/her helpful comments. The comments of the referee are in bold, our answers in normal black and the changes that have been brought to the manuscript are in blue.

**1.1 General comments**

**This manuscript presents new numerical modeling approaches to represent vertical transport of pollutants plumes in the upper troposphere with Eulerian Chemistry Transport Models (CTM). The aim is to limit the excessive vertical diffusion of the plumes of pollutant in this kind of numerical representation. Different numerical strategies are considered to address this issue : different vertical wind diagnosis, different advection scheme and different vertical resolution. The sensitivity of the simulation of a plume transport event to the different numerical choices considered in this work is evaluated on the case of the Mount Etna's eruption of March 18, 2012 Its a topic of scientific interest and certainly within the scope of Geoscientific Model Development. The general presentation of the work is logically and clearly organized. However the added value of this work could be improved with the clarification and/or the development of some results.**

**1.2 Specific Comments**

**Section 2.1 p4 l7 and l13: Could the authors precise which CHIMERE version has been used? 2013, 2017 or 2016?**

Version has been added in the title: *CHIMERE model (v2017r4; Menut et al., 2013; Mailler et al., 2017)*

**p4 l14 : The horizontal resolution of the WRF simulation should be mentioned.**

This precision has been brought to section 2.1: The horizontal grid is the same as the CHIMERE grid, with a 5 km resolution.

**p4 l17 : The authors should provide the limits of the vertical layers (at least in supplement with figure S4).**

We agree that some precision was lacking of the model vertical coordinate. However, it would be ewtremely tedious to provide all the vertical levels, and

these are not directly human-readable since this a hybrid sigma-pressure coordinate. We have added a brief description of the vertical discretization and refer the reader to the publication where the detail of the discretization strategy is provided:

The discretisation of the vertical levels is as described in [**?**], with vertical levels of exponentially increasing thickness from surface to 850 hPa, and evenly spaced (in pressure coordinates) from 850 hPa. The vertical coordinate depends on the ground-level pressure, with finer vertical levels over elevated ground. The reader is referred to [**?**] (Section 3.1) for the detailed description of the vertical discretization of the CHIMERE model.

**p4 l16-17 : There is an in-depth discussion on the relationship between horizontal resolution and vertical resolution in the article by Zhuang et al. (2018) that the authors cite in the introduction, but nothing is said here on this subject.**

A discussion of this aspect is already proposed in the introduction, though not bringing it to the same level as in Zhuang et al. 2018:

p3l8 : *Apart from this wind-mass inconsistency issue, and more specifically for the representation of polluted plumes that are transported over a long range, zhuang et al., (2018) have shown that correct representation of long-range transport of polluted plumes in the free troposphere is severely limited by the insufficient vertical resolution. They show, through dimensional and theoretical arguments, that if $\Delta x$ is not at least several hundred times $\Delta z$, representation of long-range transport of plumes in the free troposphere is hindered primarily by this coarse vertical resolution, and increasing horizontal resolution does not bring substantial added value in terms of reducing numerical diffusion of the plume. Since the $\frac{\Delta x}{\Delta z}$ in typical chemistry-transport models is around or below 20 (with a horizontal resolution of, e.g., 20 km for continental scale studies and vertical resolution of, e.g., 1 km), these authors claim that no major improvement will be reached in the representation of long-range transport plumes unless vertical resolution is refined drastically compared to current typical configurations.*

In the revised version, the reader is explicitly redirected to that study for a more in-depth discussion of this matter:

For a more detailed discussion of the theoretical ground of this relationship between horizontal and vertical discussion, the reader is referred to Zhuang et al., 2018.

**Beyond this point, it is weird to see that the chosen vertical extension of the domain does not provide any possibility to reproduce the highest part of the plume as seen by the observations (cf. figure 2e, S3 and S4). All the more so when we see that the meteorological simulation would allow the domain to be extended. Could the authors explain how they chose the different vertical resolutions tested?**

The point made by the Referee is a good point. Our choices were to guided by the idea of choosnig typical configurations for chemistry-transport models, including their drawbacks. The Chimere model is not equipped with stratospheric chemistry, and therefore 150hPa is the highest model top value that

can be chosen in the model for realistic simulations. Here of course model top could have been extended further up for the need of this particular study since we use inert chemistry, but we have the feeling that leaving the model top at 150hPa permits us to expose more of the problems that occur in typical use of chemistry-transport models, including the discussion of leakage through the model top. This matter is relevant for the simulation of long-range advection in such models, avoiding leakage of the plume through model top, but also for operational air quality forecast since, as has been shown by Emery et al., (2011), input of stratospheric ozone into the model through spurious mass fluxes at model top significantly affects operational forecast, as discussed in the introduction.

This limitation is explicitly discussed in the revised version:

Section 2.1, The top of model is placed at 150 hPa, with either 20, 50 or 99 vertical layers to evaluate the impact of vertical resolution on the volcanic plume. Even though a higher model top would have been useful for the study of this plume, 150 hPa is a typical value of top of model for CTMs that do not include stratospheric chemistry as it is the case of the CHIMERE model. Also, this relatively low value for top of model permits to examine the question of spurious mass fluxes through the top of model which, as found by Emery et al., (2011) is of relevance not only for long-range transport but also for ozone forecast to ground level.

**p4 l17 : These different vertical resolutions rely on an oversampling of the same simulated meteorological fields. Could we expect to get significantly different vertical profile with a meteorological simulation carried out with a finer vertical grid?**

Our feeling is that the scale of the vertical wind gradients in the free troposphere (a few thousand meters) is larger than the scale of the change in tracer concentration in a volcanic plume (a few hundred meters because, as discussed in, e.g., Zhang et al 2017, Eastham et al. 2018, these plumes are maintained extremely thin due to the persistant effect of wind sheer). However, we are not able to bring forward a proof of this qualitative argument, and to our knowledge a systematic evaluation of the impact of the vertical resolution of the meteorologic simulation on plume advection in chemistry-transport models is yet to be done.

**p4 l17 : Only part of the 33 vertical levels of the meteorological grid is used for the interpolation on the dispersion grid. The number of levels concerned could be specified in this paragraph.**

WRF vertical grid has been added to Figure S4 so that the reader can visualize by himself the WRF vertical discretization at the side of the CHIMERE discretization. Also, in Section 2.1, we precise that (28 levels are into 1013-150 hPa range).

**p10 l11 and p12 l5: The comparison between the different vertical resolutions involve an aspect which may deserve a bit more detailed discussion. Which kind of boundary conditions are applied for the pollutant concentrations?**

We do not have boundary condition for volcanic $SO_2$. This clarification has

[Figure]

**Figure S4.** Center : Maximum concentration altitude evolution (IASI and CHIMERE), IASI brackets indicate values' uncertainties and CHIMERE brackets indicates cell's bottom and top. Right : Model vertical levels distribution for the 3 configurations from surface to top.

been added in the revised version (Section 2.1) :

No boundary conditions were used for $SO_2$ in our simulations.

We feel that this choice is justified because we are interested in the volcanic plume only. We do not simulate the background $SO_2$ levels. If we would have made the choice to simulate these background levels, then not only an appropriate boundary condition would have been needed but also a proper set of anthropogenic emissions, which was not the purpose of the present study.

**With a plume injection in the last layer of the model (at least when 20 levels are used) it seems that the boundary conditions could play a role. What happens when the flux is downward oriented? (here again the choice of a larger vertical extension would be more relevant).**

Because there are no boundary conditions used (or, equivalently, the influx of air into the simulation domain has no $SO_2$ content), mass that is lost through upper boundary can not be brought back into it if wind turns downward. This is an issue in the 20 vertical levels cases, to a lesser extent to 50 vertical levels cases, compared to 99 vertical resolution cases, where little mass is lost through model upper boundary (q.v. Figure 3).

**p13 l1-4 : Could the authors provide the levels concerned in these tests?**

The tests have been done one the 3 vertical resolutions. It had no impact on the 20 vertical levels resolution emissions, as eruption profile width was thinner than CHIMERE top level. Only in the 99 vertical level case was observed a slight difference but not really significant on plume trajectory. It has been specified in the document : p13 l14 The tests have been conducted on 20, 50 and 99 vertical levels resolution.

**p13 l5-7 : The authors mention an "injection to a unique altitude".**

[Figure]

**Figure 3.** SO$_2$ mass evolution in model domain (kilotons). Line color indicates the vertical levels configuration, thickness indicates the vertical wind strategy considered. Dotted line represents the cumulated SO$_2$ mass emitted during Etna volcanic Eruption.

**It implies the different simulations with the different vertical resolution do not start with the same vertical extension of the plume. It would be interesting to isolate the impact of this initial discrepancy that cannot be associated to an excessive diffusion of the advection scheme. I guess this could be done with a 50 or 99 levels simulations run with an injection uniformly distributed over the different layers corresponding to the injection layer of the 20 levels simulation.**

It is possible to see the initial vertical extension of the plume on Figure 7, and indeed, simulations do start with different vertical extension of the plume.

We agree that the point brought to our attention by the Reviewer was deserving additional work. We have performed new simulations with a similar injection profile in all simulations, as the reviewer suggests, and we provide the results of these simulations in Figure S5 of the revised manuscript. This new set of simulations permits to have a better quantitative feeling of the results since avoiding the unnatural offset between the different volume curves. We are particularly grateful to the Reviewer for this suggestion.

A paragraph has been added in the manuscript to describe the results :

To evaluate the impact schemes and vertical resolution would have with a similar vertical extension at injection, new simulations have been conducted imposing an identical vertical distribution at the first time (spreading verticaly the emited mass over the same thickness in the 50 and 99-level simulations than it has in the 20-level simulation). Simulations have been conducted for 20, 50 and 99 vertical levels, for WRFW-DL and NODIV-VL parameters, a total of six simulations. Results have been displayed in supplements, on Figure S5. It can be seen on Figure S5 (left) that all plumes have the same initial volume regardless of vertical resolution , which was not the case in the previous case (c.f. Figure 8a). With a larger vertical extension of the plume at injection,

volumes are higher than in the "unique cell injection" cases, but resolution and transport scheme influence in the same way the evolution of plume (considering its volume). Figure S5 (right) shows evolutions of $SO_2$ highest column vertical profile, similar to Figure 7. This new set of experiments show that, even when getting rid of the initial distorsion due to sharper injection profiles in the simulations with the most refined vertical distributions, the increase in plume volume is much slower in the 99-level simulations than in the 20-level simulations. The final volume is about 4 times smaller in the 99-level simulations compared to their 20-level counterparts. A similar factor in vlume reduction is obtained by changing strategy from VL-NODIV to DL-REALW. In total, final plume volume in the worst-case NODIV-VL-20 simulation is about 20 times bigger than final plume volume in the best-case WRFW-DL-99 simulation.

**p14 section 3.3 : With the location of "the model column with the strongest vertically integrated SO2 content" the authors have chosen a very aggregated indicator for the comparison between satellite soundings and model results. I assume this choice was made for sake of simplicity in the presentation of the results. However, seeing that the configuration option can lead to some plume splitting, it would be interesting to have more information concerning the horizontal extension of the plume in the different cases.**

The aim of section 3.6 *Parameters impact on $SO_2$ dispersion* is to evaluate plume diffusion over 3 dimensions (minimum volume containing 50 % of the $SO_2$ mass), and volume results are applicable to surface (cf. p19 l7: *By extension, it has been observed that volcanic plume shape has been modified by DL and WRFW parameters, reducing the surface area containing 50 % of $SO_2$ total mass*).

To illustrate the differences, Figure S6 (in suppl.) has been added to show the horizontal dispersion of plume on various simulations after 2 days.

Also, information on the horizontal area of the plume has been added (Fig. S7) and commented briefly in the manuscript: We have also calculated the minimum area containing more than 50% of the $SO_2$ mass (Fig. S7), showing that the WRFW-DL simulations concentrate 50% of the plume mass in an area at least twice as small as their NODIV-VL counterparts.

**p14 l23 : It is not clear to me if the results in figure 5 present average over different configuration options. For instance in the first panel, the simulations with the different advection scheme are compared. Do the number are averages over the different vertical resolutions (the vertical resolution is not mentioned either in the text or in the label of the figure)? Does this imply that there is few interaction between the tested options?**

To produce this figure, differences (in km) between model and satellite plumes centroids are calculated for each simulations, then parameters impact are evaluated by calculating the mean between simulation-satellite differences. For instance, "NODIV-DL" ($1^{st}$ line, left column) is the mean between "NODIV-DL-20", "NODIV-DL-50", "NODIV-DL-99". "NODIV-99" ($3^{rd}$ line, left column) is the mean between "NODIV-DL-99" and "NODIV-VL-99". This method has been used to better evaluate the impact of each parameter independently,

[Figure]

**Figure 7.** Evolution of SO$_2$ vertical profile (in ppb) corresponding to the maximum column for each step after the Etna eruption, for each tested model configurations. 1$^{st}$ row: NODIV-VL; 2$^{nd}$ row: NODIV-DL; 3$^{rd}$ row: WRFW-VL; 4$^{th}$ row: WRFW-DL. Left: 20 vertical levels; Center: 50 vertical levels; Right: 99 vertical levels. WRFW simulations values have been corrected to fit NODIV strategy masses.

[Figure]

**Figure S5.** Left) Minimum volume evolution calculated for 50 % of SO$_2$ total mass in the atmosphere. Right) Evolution of SO$_2$ vertical profile (in ppb) corresponding to the maximum column for each step after the Etna eruption, for each tested model configurations. 1$^{st}$ row: NODIV-VL; 2$^{nd}$ row: WRFW-DL. Left: 20 vertical levels; Center: 50 vertical levels; Right: 99 vertical levels. WRFW simulations values have been corrected to fit NODIV strategy masses.

[Figure]

**Figure S6.** Volcanic plume integrated column dispersion on march 20$^{th}$ at 11 A.M. UTC (2 days after the eruption).

instead of each simulation. The caption has been expended to help the reader better understand the figure:

To produce this figure, differences (in km) between model and satellite plumes centroids are calculated for each simulation, then parameters impact are evaluated by calculating the mean between simulation-satellite differences. For instance, "NODIV-DL" ($1^{st}$ line, left column) is the mean between "NODIV-DL-20", "NODIV-DL-50", "NODIV-DL-99". "NODIV-99" ($3^{rd}$ line, left column) is the mean between "NODIV-DL-99" and "NODIV-VL-99".

**From figure 4 we can see that the WRFW-DL-99 simulation is not the closest to the observation at the final stage. This may not be the intuition get from the results presented.**

We present the trajectories to explain what the more synthetic results in, e.g. Fig. 5, mean. It is almost impossible for us to visually extract the information from this set of 12 curves and sort out the effect of all three variable parameters in the simulations, this is why we chose to build more synthetic indices and average simulation ensembles together to isolate as much as possible the effect of the parameters without having too much influence of the good or bad luck that can impact every separate simulation.

On Fig. 4, "WRFW-DL-99" is among a set of, say, 4 simulations that are the closest to the observed satellite plume at final stage, but not the closest.

**p16 l18-20 : Could the authors precise how the distribution are built? It is not clear for me if it represents different time steps, different locations, or a mix? Are the observations uncertainties are represented in this figure?**

Brackets correspond to distribution's $10^{th}$ and $90^{th}$ percentiles (precision now brought to Figures's caption) and observation uncertainties are not represented in this figure: the figure represents only the spread in the satellite-retrieved altitudes and in the modelled altitudes, for the easter plume (above) and the western plume (below), c.f. Figure 2 and Table 3, column $6^{th}$ ($\lambda_{\mathrm{thr},i}$).

**p17 l6-8 : These lines are frustrating from my point of view. The authors focus their work on the excessive vertical diffusion in the dispersion model and the only comparison of the model results to observation concerning the plume vertical extension indicates that this plume property is underestimated. Could the authors provide a more in-depth discussion concerning this point? Some considerations concerning the time evolution of the maximum concentration (modeled and observed) could be useful here to convince the readers that a less diffusive treatment of the advection is really suitable.Since the transport in this application is linear, even a normalized comparison to the "initial" (sounding number 1) maximum concentration would be useful.**

We understand the frustration of the Referee about this point, it is our frustration too. However, the large "brackets" in the 10th to 90th percentiles for the satellite measurements are, unfortunately, due only to large uncertainties in the satellite retrievals. In the same line, estimates of maximal concentration in the satellite data are very uncertain, in part but not only due to the uncertainty

on the vertical profiles of the satellites. This is why we are able to use comparison to satellite data only to check the general structure of the modelled plume but unfortunately not to give a comparison point on vertical diffusion. Even though we are not able to prove it separately in the present study due to inadequacy of our satellite data for this purpose, we consider that excessive vertical diffusion in Eulerian CTMs are already a well-known and very general problem (e.g. Colette et al. 2010, Emery et al. 2011, Zhuang et al. 2018 etc.).

In the revised version, we add a sentence to explicitly state the limits of the comparison to satellite data in link with uncertainties of the latter

The dataset also provides error-range estimates along with the retrieved plume altitude. These error-range estimates have a median of around 1000 m in the western plume and 5000 m in the eastern plume, which is much higher aloft. These uncertainties help to understand the wide distribution obtained from satellite. It is also worth noting that this dataset provides pume altitude but does not provide an information on plume thickness. Therefore, comparison between the left and right panels in Figure ?? does not represent the compared plume thickness between model and observation, but the compared variability of plume height. Unfortunately, due to the relatively large ucertainties affecting the retrieved altitudes, no conclusion can be made on this point either. With all these imitations, Fig. ?? prove that model simulations represent the general structure of the pkume, with an elevated eastern plume and a low western plume, and that the median altitudes of both these plumes are very comparable to the median of the satellite-provided altitudes.

**1.3   Technical corrections**

**p5 l11 : It seems an "overbar" is missing for notation consistency.** It has been modified, thanks

**p11 label table 3 :  The last sentence should probably be in the label of Figure 2.**

It has been modified.

**p18 l21 : erroneous citation**

Després and Lagoutière (1999) changed for *Van Leer (1977).*